# Advantageous and disadvantageous inequality aversion can be taught through learning of others' preferences

Shen Zhang[1,2]*, Oriel FeldmanHall[3,4], Sébastien Hétu[5,6], A Ross Otto[7]

[1]State Key Laboratory of Cognitive Neuroscience and Learning & IDG/McGovern Institute for Brain Research, Beijing Normal University, Beijing, China; [2]Department of Neurobiology, German Primate Center, Göttingen, Germany; [3]Cognitive, Linguistics and Psychological Sciences, Brown University, Providence, United States; [4]Carney Institute for Brain Sciences, Brown University, Providence, United States; [5]Department of Psychology, Université de Montréal, Montréal, Canada; [6]Centre Interdisciplinaire de Recherche sur le Cerveau et l'Apprentissage (CIRCA), Montréal, Canada; [7]Department of Psychology, McGill University, Montréal, Canada

## eLife Assessment

This cleverly designed and potentially **important** work supports our understanding regarding how and whether social behaviours promoting egalitarianism can be learned, even when implementing these norms entails a cost for oneself. However, the evidence supporting the major claims is currently **incomplete**, with the major limitation being whether Ps truly learn egalitarianism from a teacher or instead exhibit reduced guilt across time that is reduced when observing others behaving more selfishly. With a strengthening of the supporting evidence, this work will be of interest to a wide range of fields, including cognitive psychology/neuroscience, neuroeconomics, and social psychology, as well as policy making.

*For correspondence:
shen.zhang@mail.bnu.edu.cn

Competing interest: The authors declare that no competing interests exist.

**Abstract** While enforcing egalitarian social norms is critical for human society, punishing social norm violators often incurs a cost to the self. This cost looms even larger when one can benefit from an unequal distribution of resources, a phenomenon known as advantageous inequity—for example, receiving a higher salary than a colleague with the identical role. In the Ultimatum Game, a classic testbed for fairness norm enforcement, individuals rarely reject (or punish) such unequal proposed divisions of resources because doing so entails a sacrifice of one's own benefit. Recent work has demonstrated that observing and implementing another's punitive responses to unfairness can efficiently alter the punitive preferences of an observer. It remains an open question, however, whether such contagion is powerful enough to impart advantageous inequity aversion to individuals—that is, can observing another's preferences to punish inequity result in increased enforcement of equality norms, even in the difficult case of Advantageous inequity? Using a variant of the Ultimatum Game in which participants are tasked with responding to fairness violations on behalf of another 'Teacher'—whose aversion to advantageous (versus disadvantageous) inequity was systematically manipulated—we probe whether individuals subsequently increase their punishment unfairly after experiencing fairness violations on their own behalf. In two experiments, we found individuals can acquire aversion to advantageous inequity through observing (and implementing) the Teacher's preferences. Computationally, these learning effects were best characterized by a model which learns the latent structure of the Teacher's preferences, rather than a simple Reinforcement Learning account. In summary, our study is the first to demonstrate that people can swiftly and readily acquire

another's preferences for advantageous inequity, suggesting in turn that behavioral contagion may be one promising mechanism through which social norm enforcement—which people rarely implement in the case of advantageous inequality—can be enhanced.

## Introduction

Humans can learn how to navigate through the world by observing the actions of others. For example, individuals can learn complex motor skills by observing and imitating how experts coordinate their movements (*Hayes et al., 2008*). Observational learning can also transmit important information about social and moral norms, such as reciprocity (*Engelmann and Fischbacher, 2009*), cooperating with others (*van Baar et al., 2019*), or context in which punishment is considered appropriate (*FeldmanHall et al., 2018*). In some cases, an individual learns from others in a straightforward and unambiguous social context, where the tensions endemic to many moral dilemmas—for example, benefit oneself versus the collective good—are not directly juxtaposed against one another. And yet, the social world is rarely straightforward, often ambiguous, and moral dilemmas that pit self-benefit over the collective good abound (*FeldmanHall and Shenhav, 2019*; *Vives and FeldmanHall, 2018*).

Consider the case of inequity, for which individuals exhibit a strong distaste: concerns for fairness are well documented in adults (*Güth et al., 1982*; *Sanfey et al., 2003*), children (*Fehr et al., 2008*; *McAuliffe and Dunham, 2017*), primates (*Brosnan and De Waal, 2003*; *Brosnan and de Waal, 2014*; *van Wolkenten et al., 2007*), and even domesticated dogs (*Essler et al., 2017*). This aversion to inequity manifests perhaps most famously in the Ultimatum Game, in which one player (the Proposer) decides how to split a sum of money with another player (the Receiver), a role typically assumed by the participant (*Güth et al., 1982*; *Sanfey et al., 2003*). A Receiver's acceptance results in both parties receiving the offered money, whereas rejection results in neither party receiving any money—a form of costly punishment. A recurring observation supporting the notion of inequity aversion is that people tend to reject disadvantageous offers that unfairly benefit the other party (*Brosnan and de Waal, 2014*; *Fehr and Schmidt, 1999*).

At the same time, not all inequity is experienced in the same way. For example, when we stand to receive less than our 'fair share', such disadvantageous inequity (Dis-I for short) engenders feelings of envy, anger, and/or disappointment (*Heffner and FeldmanHall, 2022*) which often manifests in punishment of unfair offers in the Ultimatum Game via rejection (*McAuliffe et al., 2014*; *McAuliffe et al., 2017*; *Pedersen et al., 2013*; *Pillutla and Murnighan, 1996*). In contrast, when we stand to receive a favorable (albeit unfairly distributed) share of resources, these advantageous inequitable (Adv-I) offers often engender feelings of guilt or shame (*Gao et al., 2018*). Despite these negative emotions, Receivers in Ultimatum Game settings are much less willing to engage in costly punishment of offers that are advantageously inequitable (*Civai et al., 2012*; *Hennig-Schmidt et al., 2008*; *Luo et al., 2018*). In short, Adv-I versus Dis-I engender markedly different punishment preferences.

A growing body of developmental research demonstrates that this difference in punitive responses to Adv-I versus Dis-I manifests early in the developmental trajectory (*Amir et al., 2023*; *Blake et al., 2015*; *Blake and McAuliffe, 2011*; *McAuliffe et al., 2017*). Indeed, punishing Adv-I offers requires sacrificing a (larger) personal gain to achieve a fairer outcome, mirroring many moral dilemmas in which self- versus other-regarding interests are at odds. Because aversion to Adv-I is thought to arise from more abstract concerns about fairness (*Tomasello, 2019*), Adv-I aversion is thought to impose considerable demands on more sophisticated cognitive processing (*Gao et al., 2018*). This may explain, in part, why the developmental trajectory of Adv-I-averse preferences comes online much later relative to Dis-I (*McAuliffe et al., 2017*).

Thus, one open question concerns how people acquire inequity-averse preferences. One influential framework posits that we often adapt our behaviors to people around us through a process of conformity (*Cialdini and Goldstein, 2004*). Put simply, loyally following the behaviors of another—a social contagion effect—is a powerful motivator of social behavior. Such behavioral contagion effects are observed in diverse decision-making domains such as valuation (*Campbell-Meiklejohn et al., 2010*), risk-taking (*Suzuki et al., 2016*), delay of gratification (*Garvert et al., 2015*), moral preferences (*Bandura and McDonald, 1963*; *Vives et al., 2022*), and social norms (*Hertz, 2021*). Recently, we demonstrated that punitive responses to Dis-I can be 'taught' to participants in the context of the Ultimatum Game, whereby individuals' preferences for rejecting disadvantageous unfair offers were

strengthened as a result of observing another individual's (a 'Teacher') desire to punish these offers (*FeldmanHall et al., 2018*).

At the same time, social contagion effects may be far less robust if the behavior demands sacrifice of self-benefit. Since this type of dynamic places behavioral contagion and desire for self-gain in opposition, it remains unclear whether Adv-I aversion can also be acquired 'vicariously' through observing another's preferences. Here, we investigate whether the act of observing the Adv-I-averse preferences of another punitive Receiver enhances an individual's aversion to Adv-I, even if the rejection of such Adv-I offers requires sacrificing self-benefit. Answering this question not only enriches our understanding of the nature of inequity aversion, but also enables us to better understand the mechanisms underpinning learning of others' moral preferences during social interactions.

Computationally, one can imagine different possible mechanisms driving learning of inequity aversion. One possibility is that observational learning of moral preference is based on simple action-outcome contingencies. On this view, a simple but elegant Reinforcement Learning (RL; *Burke et al., 2010*; *Diaconescu et al., 2020*; *FeldmanHall et al., 2018*; *Lindström et al., 2019*) model formalizes how individuals adapt their behavior to recently observed outcomes. In its most basic form, RL makes choices on the basis of learned associations between actions and outcomes, and critically, actions are bound to the specific decision context—in the case of the Ultimatum Game, the unfairness of the amount offered by the Proposer. However, during social interactions, people might not consider the behaviors of others as resulting from simple action-outcome associations but alternatively, construct and use models of other agents, representing their stable intentions, beliefs, and preferences (*Anzellotti and Young, 2020*). Accordingly, we also consider the possibility that moral preferences are immutable across contexts (*Bail et al., 2018*; *Fehr and Schmidt, 1999*; *Taber and Lodge, 2012*), which would suggest that moral preferences are not learned merely as associations (i.e. specific responses tied to different unfairness levels), but rather, through a deeper inference process which models the underlying fairness preferences of the observed individual. Here, across two experiments, we build upon a well-characterized observational learning paradigm (*FeldmanHall et al., 2018*; *Son et al., 2019*; *Vives et al., 2022*) to examine the conditions under which individuals are able to learn Adv-I-averse preferences on the basis of exposure to another Receiver's punitive preferences. In addition, to mechanistically probe how punitive preferences may be acquired in Adv-I and Dis-I contexts we also characterize trial-by-trial acquisition of punitive behavior with computational models of choice.

## Results
### Experiment 1

In Experiment 1, following the approach of previous experiments (*FeldmanHall et al., 2018*), we test if the rejection of advantageous unfair offers can be learned on the basis of exposure to the preferences of another individual (the 'Teacher') exhibiting Adv-I-averse preferences. In a between-subject design with three phases, participants interacted with other individuals in a repeated Ultimatum Games (*Figure 1a*). We assess contagion effects by measuring participants' Dis-I and Adv-I aversion both before and after observing (and implementing) the preferences of a Teacher who exhibits inequity aversion in both Dis-I and Adv-I contexts ('Adv-Dis-I-Averse' condition; N=100) and inequity aversion only in a Dis-I context ('Dis-I-Averse' condition; N=100).

First, to assess participants' baseline fairness preferences across inequity levels, in the Baseline Phase (*Figure 1c*) participants acted as a Receiver in several one-shot UGs, responding to offers ranging from extreme Dis-I (e.g. the Proposer keeps 90 cents and offers 10 cents to the Receiver; a 90:10 split) to extreme Adv-I (e.g. the Proposer keeps 10 cents and offers 90 cents to the Receiver; a 10:90 split) in a total of five offer types (90:10, 70:30, 50:50, 30:70, 10:90). On each trial, participants interacted with a different Proposer, and unbeknownst to participants, the offers were pre-determined by the experimenters. Following the typical formulation of the UG (*Güth et al., 1982*), participants made the choice between accepting versus rejecting each offer and also rated the fairness of the offer.

Next, in the Learning Phase (*Figure 1d*), participants played a repeated UG as a third party rather than a Receiver, in which they accepted or rejected offers on behalf of another Receiver (termed the Teacher in this phase) such that the participant's decisions did not impact their own payoff but would impact the payoffs to the Proposer and the Teacher. Critically, after each decision, participants received

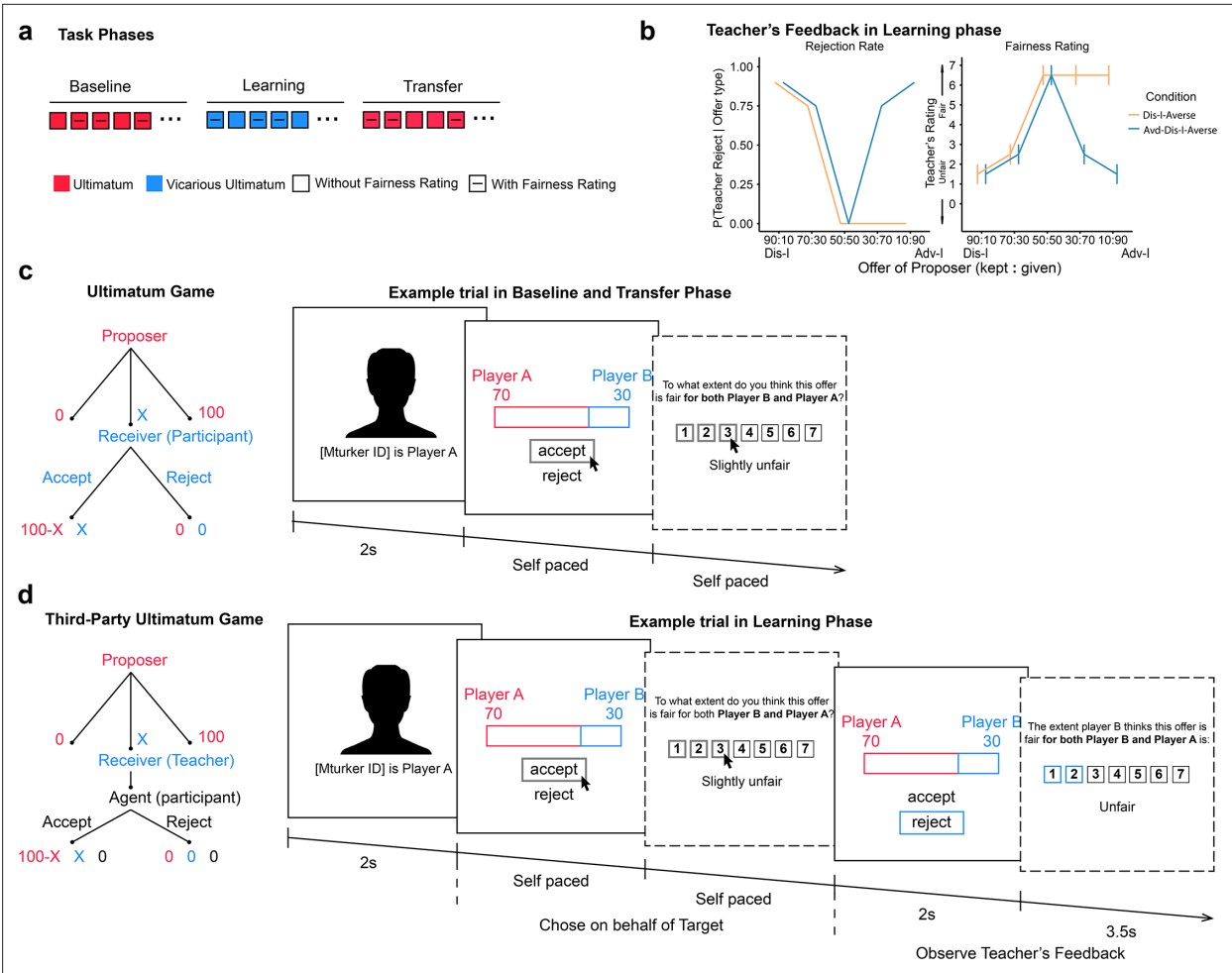

**Figure 1.** Experiment procedure and task design. (**a**) Task Overview. Our main task consists of three phases. In the Baseline Phase, participants acted as a Receiver, responding to offers of different inequity levels and rated their perceived fairness of the offers on three out of every five trials. In the subsequent Learning Phase, participants acted as an Agent, deciding on behalf of the Receiver (Teacher) and Proposer. Again, they rated the fairness on three out of every five trials. Finally, participants made choices in a Transfer Phase which was identical to the Baseline Phase. (**b**) Preferences and Fairness Ratings governing the Teacher's feedback in the Learning Phase (See *Supplementary file 1A* and *Supplementary file 1B*). (**c**) Baseline and Transfer phase, in which participants played the Ultimatum game as a Receiver, making choices on their own behalf. (**d**) In the Learning phase, participants acted as a third party (the agent), making decisions on behalf of the Proposer and the Receiver (Teacher), playing a Third-Party Ultimatum game. In a Third-Party Ultimatum game, the participant makes decisions on behalf of the Receiver: if they rejects the proposed split, both the Proposer and the Receiver receive nothing. If they accept, the Proposer and the Receiver are rewarded with the proposed split.

feedback whether the Teachers *would* have preferred acceptance versus rejection (i.e. punishment) of the offer. Thus, through trial-by-trial feedback, the Teacher can signal to participants their preference to punish the Proposer for making unfair offers. Critically, in the 'Dis-I-Averse' condition, akin to the typical pattern of human preferences observed in the UG (*Güth et al., 1982*; *Sanfey et al., 2003*), the Teacher's preferences exhibited a strong aversion to Dis-I, and thus routinely punishing unfair offers. Specifically, the Teacher was likely to reject Dis-I offers (i.e. 90:10 and 70:30), but not Adv-I offers (i.e. 30:70 and 10:90; see *Figure 1b* and *Supplementary file 1A*). However, in the 'Adv-Dis-I-Averse' condition, the Teacher was likely to reject any unfair offer, regardless of whether it was Dis-I or Adv-I, manifesting typical Dis-I-averse preferences as well as the less commonly observed aversion to Adv-I (i.e. punishing advantageous offers). Feedback from the Teacher was also accompanied by fairness ratings consistent with their preferences (see *Figure 1b* and *Supplementary file 1B*).

Finally, to examine contagion (or transmission) of the Teacher's preferences to the participants, we assessed fairness preferences of the participants for a second time in a Transfer Phase (*Figure 1a*).

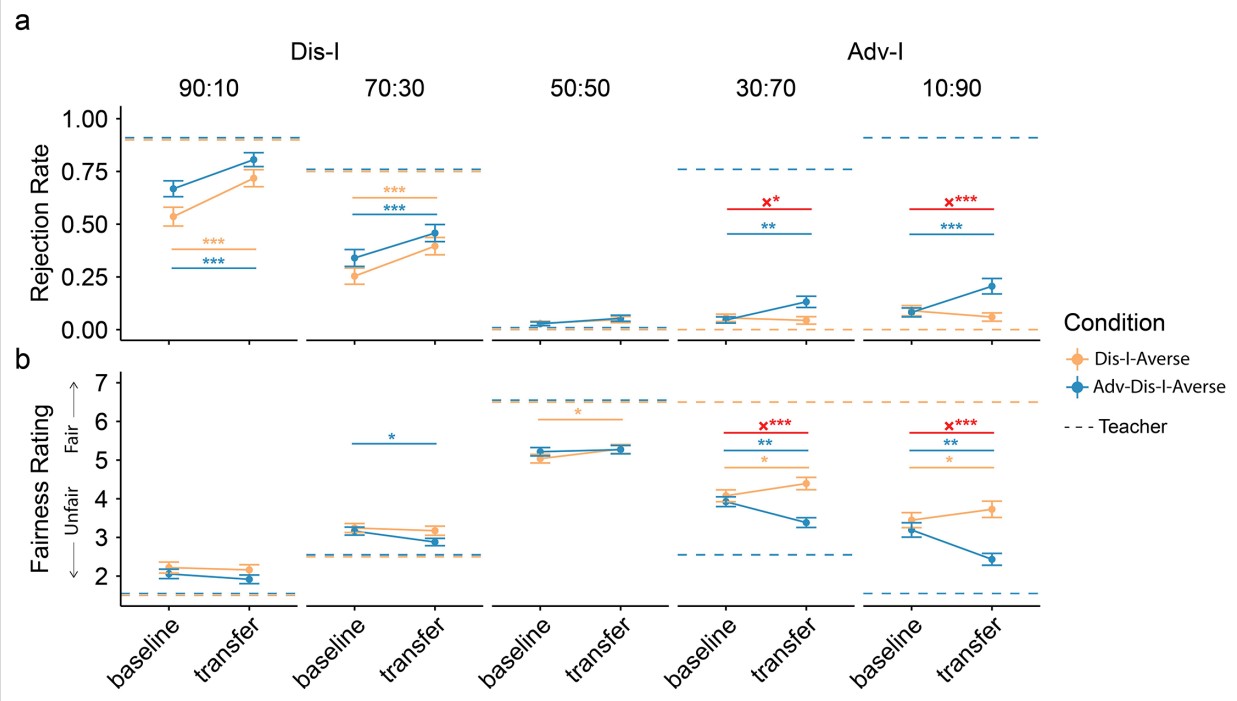

**Figure 2.** Behavioral contagion in Experiment 1. (**a**) Rejection rates change significantly in Dis-I offers for both learning conditions, while changes in Adv-I offers were only evident in Adv-Dis-I-Averse condition. (**b**) Participants in the Adv-Dis-I-Averse condition exhibited significant changes in fairness ratings, and these fairness rating changes differed significantly between conditions. Dashed lines indicate punishment preferences (Panel a) and fairness ratings (Panel b) exhibited by the Teacher . Error bars indicate standard error (†indicates p<0.1, *indicates p<0.05, **indicates p<0.01, ***indicates p<0.001,×indicates interaction). Resutls from linear mixed models (LMM, n = 100 for both conditions).

This third phase was identical to the Baseline Phase and thus allowed us to quantify changes in participants' fairness preferences before and after the Learning Phase.

### Preferences across baseline and transfer phases

Mirroring preferences typically observed in Western, Educated, Industrialized, Rich, and Democratic (WEIRD) participant populations (**Henrich et al., 2006**), punishment choices in the Baseline Phase were Dis-I-averse, but not Adv-I-averse, as we observed similarly high rejection rates during the Baseline Phase for Dis-I offers across the Dis-I-Averse and the Adv-Dis-I-Averse conditions (**Figure 2**, **Supplementary file 1C**; all ps < 0.001). As expected, the rejection rates for Adv-I offers were much lower than Dis-I offers (linear contrasts; e.g. 90:10 vs 10:90, all ps < 0.001, see **Supplementary file 1C** for logistic regression coefficients for rejection rates). Consistent with these Rejection rates, participants rated Dis-I offers as unfair (i.e. lower than the rating scale midpoint of 4, **Supplementary file 1D**; all ps < 0.001) and Adv-I offers were rated as more fair than Dis-I offers (all ps < 0.001)—despite the fact that the offers in the Adv-I and Dis-I contexts represent the same magnitude of inequity (e.g. 90:10 vs 10: 90 splits).

To examine whether exposure to the Teacher's punishment preferences in the Learning Phase resulted in changes to participants'preferences, we examined changes in Rejection rates and Fairness ratings between the Baseline and Transfer phases (**Figure 2**). Overall, we found robust changes in Rejection rates and Fairness ratings between the Baseline and Transfer phases (**Figure 2**, **Supplementary file 1E, F**). Importantly, when comparing these changes between the two learning conditions, we observed significant differences in rejection rates for Adv-I offers: compared to exposure to a Teacher who rejected only Dis-I offers, participants exposed to a Teacher who rejected both Dis-I and Adv-I offers were more likely to reject Adv-I offers and rated these offers as more unfair. This difference between conditions was evident for both 30:70 offers (Rejection rates: $\beta$(SE)=0.10(0.04), p=0.013; Fairness ratings: $\beta$(SE)=−0.86(0.17), p<0.001) and 10:90 offers (Rejection rates: $\beta$(SE)=0.15(0.04), p<0.001, Fairness ratings: $\beta$(SE)=−1.04(0.17), p<0.001). As a control, we also compared rejection rates

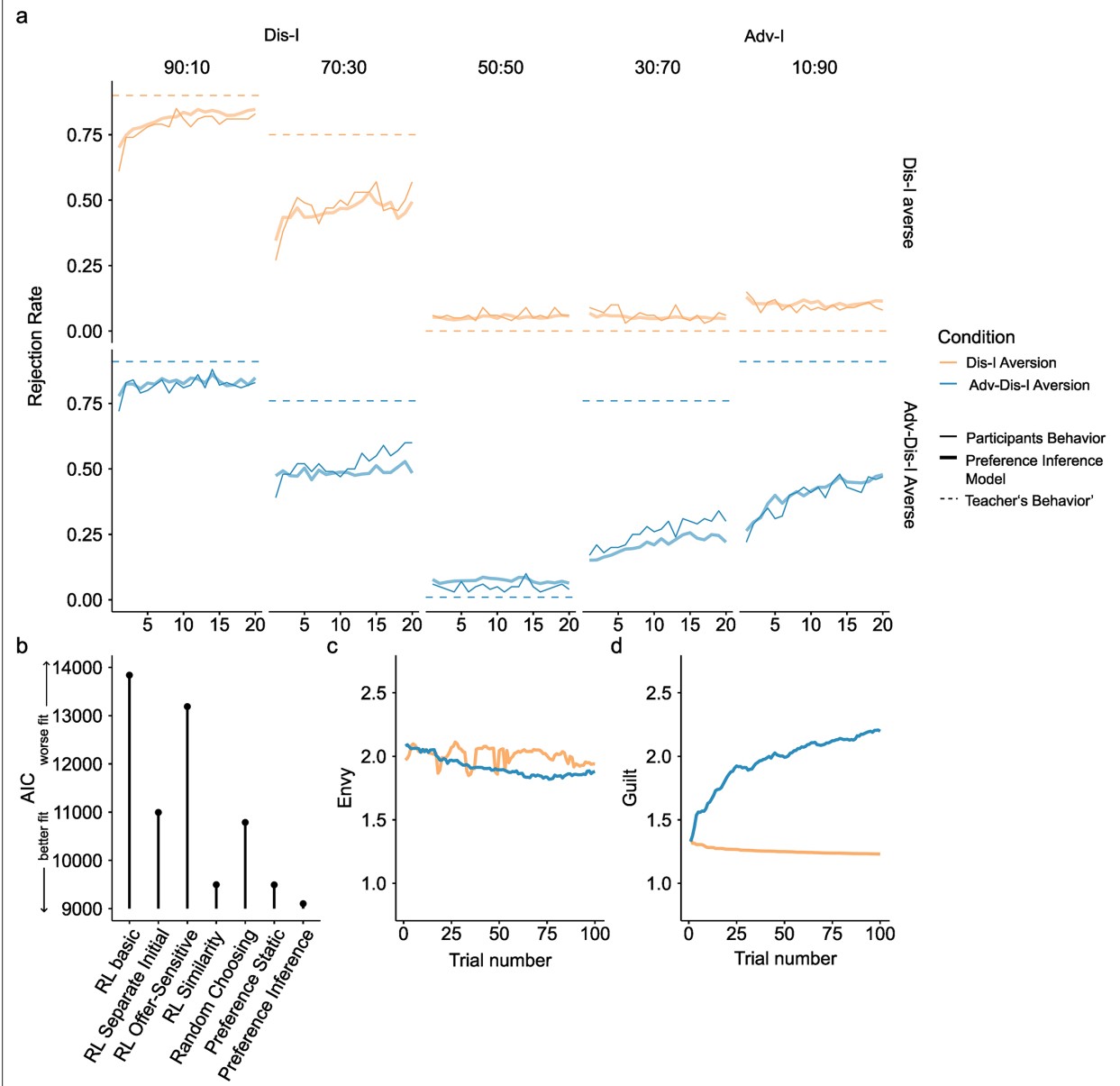

**Figure 3.** Learning phase behavior in Experiment 1. (**a**) Rejection rate changes in the Learning Phase. For Dis-I Offers, rejection rates increased for both learning conditions, while rejection rates only increased for Adv-I offers in the Adv-Dis-I-Averse condition. Furthermore, in Adv-I offers, the increasing trend was larger in the Adv-Dis-I-Averse condition than in the Dis-I-averse condition, indicating a learning effect. Solid thin lines denote participants' rejection choices, dashed lines denote the Teacher's preferences, and solid thick lines represent predictions of the (best-fitting) Preference Inference Model. (**b**) Model comparison demonstrating that the Preference Inference model provided the best fit to participants' Learning Phase behavior (AIC: Akaike Information Criterion) (**c and d**). Parameter learning for the Preference Inference model, which captured a significant rejection rate increase in Adv-I offers by updating the guilt parameter in a trial-by-trial manner.

The online version of this article includes the following figure supplement(s) for figure 3:

**Figure supplement 1.** Model recovery results for the Preference Inference model.

and fairness rating changes between conditions in Dis-I offers (90:10 and 30:70) and Fair offers (i.e. 50:50) but observed no significant difference (all ps >0.217).

## Learning another's preferences
Having demonstrated that participants' preferences to reject unfair Adv-I offers were altered on the basis of exposure to the Teacher's preferences, we next examined trial-by-trial changes in rejection

rates during the Learning phase (*Figure 3a*). A mixed-effects logistic regression revealed a significantly larger (positive) effect of trial number upon rejection rates for Adv-I offers in the Adv-Dis-I-Averse condition compared to the Dis-I-Averse condition. This relative rejection rate increase was evident both in 30:70 offers (*Supplementary file 1G*; $\beta(SE)=0.77(0.24)$, p<0.001) and in 10:90 offers ($\beta(SE)=1.10(0.33)$, p<0.001). In contrast, comparing Dis-I and Fairness offers when the Teacher showed the same tendency to reject, we found no significant difference between the two conditions (90:10 splits:$\beta(SE)=-0.48(0.21)$, p=0.593 ; 70:30 splits: $\beta(SE)=-0.01(0.14)$, p=0.150; 50:50 splits: $\beta(SE)=-0.00(0.21)$, p=0.086). In other words, participants by and large appeared to adjust their rejection choices in accordance with the Teacher's feedback in an incremental fashion.

## Computational models of learning punishment preferences

Having established that rejection rates increased for both Dis-I offers (in both the Dis-I-Averse and Adv-Dis-I-Averse conditions) and Adv-I offers (only in the Adv-Dis-I-Averse condition), we then sought to better understand the learning mechanisms underpinning trial-by-trial learning (*Figure 3a*). We used computational modeling to formalize two different sets of assumptions regarding how participants learn from Teachers' feedback. Under one account, a simple RL model proposes that decision-makers learn punishment preferences by observing feedback resulting from actions made in response to specific offers. Previously, we have found that this 'naive' model provided a reasonable characterization of participants' trial-by-trial learning of Dis-I-averse preferences (*FeldmanHall et al., 2018*). However, this model may fail to capture a critical facet of learning: participants' moral preferences may not be learned merely as associations—the type of response being tied to specific offers—but rather, through a deeper inference process which models the underlying fairness preferences of the Teacher. Accordingly, our alternative model assumes that participants use trial-by-trial feedback to infer the Teacher's underlying preferences concerning inequality, which may shift depending on the context (Dis-I versus Adv-I). Following the simple Fehr-Schmidt inequality aversion formalism *Fehr and Schmidt, 1999*, in our model—termed the Preference Inference model—the Teacher's aversion to Dis-I is modeled by an 'Envy' parameter, while the 'Guilt' parameter captures the Teacher's aversion to Adv-I (see Materials and methods). Critically, the RL model does not learn the Teacher's preferences per se, but the value of each action (accept or reject), independently for each offer type. In contrast, the Preference Inference model explicitly represents the extent of the Teacher's Adv-I- and Dis-I-aversion—that is their underlying preferences—by independently updating the envy and guilt parameters using trial-by-trial feedback from the Teacher.

We compared the goodness of fit of seven different models to participants' choices in the Learning Phase: the Preference Inference model, which learns the 'guilt' and 'envy' parameters experientially, a Static Preference model that assumes the 'guilt' and 'envy' are fixed over the course of learning (baseline model 1, see Materials and methods), three variants of a simple RL model that make different assumptions about how action values are represented (models 3-5), and a baseline or 'null' model that assumes a fixed probability of each action (randomly choosing, baseline model 2). While simplistic, RL models were able to characterize observational learning of punishment successfully in our previous study (*FeldmanHall et al., 2018*) and thus serve as a reasonable baseline for evaluating more sophisticated models. Finally, we explored the possibility that a simple RL model imbued with the ability to generalize learning between similar offer types could capture Learning Phase behavior (the Similarity RL model; see Materials and methods). We fit each of the seven models via maximum likelihood estimation, penalizing for model complexity, and found that the Preference Inference model provided the best characterization of learning (*Figure 3b*, *Supplementary file 1H*), suggesting that participants were performing trial-by-trial inference of the Teacher's underlying inequity preferences, rather than simply learning reinforced associations between experienced offer types and actions. Even the Static Preference model, which does not assume any learning mechanism (but rather assumes fixed preferences with respect to Dis-I and Adv-I) provided a better characterization of learning than any of the three naive RL models which do assume incremental learning over paired associations. Furthermore, the Preference Inference model also outperforms the Similarity RL, which assumes a generalization mechanism.

To examine the learning dynamics underpinning the best-fitting Preference Inference Model, we simulated Learning Phase choice behavior using participants' estimated parameter values (*Figure 3a*, see Materials and methods for details). The close correspondence between the simulated and

observed learning curves indicates that the Preference Inference model captures the reinforcement-guided variations in punishment in Dis-I-context and, crucially, the marked differences in learning between Dis-I-Averse and Adv-Dis-I-Averse conditions in Adv-I-context (30:70 and 10:90 splits). To better understand how the Preference Inference model accounts for these patterns of change in rejection rates, we examined how model-inferred aversion to Dis-I ('envy') and Adv-I ('guilt')—the two components of the Fehr-Schmidt inequality aversion model (*Fehr and Schmidt, 1999*) representing the latent structure of the Teachers' preferences—emerge as a function of exposure to the Teacher's preferences. In both the Dis-I-Averse and Adv-Dis-I-Averse conditions, the model inferred a similarly 'envious' Teacher (*Figure 3c*), while the model only increased its estimate of the Teacher's 'guilt' parameter (*Figure 3d*) in the Adv-Dis-I-Averse condition (wherein the Teacher's feedback manifested Adv-I aversion) mirroring the model's—and participants'—shift in rejection rates over the course of the Learning Phase and further suggesting that the Preference Inference model captured critical aspects of participants' learning of the Teachers' preferences.

Experiment 1 extends our previous work (*FeldmanHall et al., 2018*) by revealing that Adv-I-averse preferences—which are believed to be less mutable (*Luo et al., 2018*)—can be similarly shaped by exposure to another individual with manifesting a strong aversion to resource divisions that unfairly benefit them. These results suggest that individuals' moral preferences can be learned even in cases where these preferences conflict with one's own self-interest. Computationally, this learning process was best characterized by an account that prescribes that individuals build a representation of others' moral preferences about Dis-I and Adv-I (akin to the *Fehr and Schmidt, 1999* model of inequity aversion), rather than by a simple RL account. This preference inference account predicts that individuals, when exposed to only a fraction of inequity-related punishment preferences, should generalize these inferred preferences to other similar inequity contexts. In Experiment 2, we sought to test this generalization hypothesis more directly, buttressing the idea that the learning and transfer of Inequity-averse preferences observed in Experiment 1 came about as a result of participants modeling the Teacher's inequality-averse preferences.

## Experiment 2

Experiment 2 provides a more stringent test of whether participants model the Teacher's underlying inequity preferences. Specifically, if individuals indeed learn the Teacher's latent inequity-averse preferences in the Learning phase, we would expect that feedback-driven learning of Teacher's punishment preferences on specific (moderate inequity) offers (30:70 splits) should generalize to offers in the same context (10:90 splits) without any direct experience of those offer types. Accordingly, to probe for this sort of generalization—a hallmark of the sort of latent structure learning we attribute to the behavior we observed in Experiment 1—we now eliminate feedback for extreme Dis-I (90:10) and Adv-I (10:90) offers from the Learning phase. If participants' punishment preferences are informed by modeling inferred inequity-averse preferences of the Teacher, we should expect to see these preferences transfer to participants' own fairness preferences in a similar, generalized manner in the Transfer Phase.

Mirroring Experiment 1, we employed two conditions governing the Teachers' preferences in the Learning Phase: In the Dis-I-Averse condition, the Teacher only exhibited strong punishment preferences concerning moderate Dis-I offers (30:70 splits), while the Teacher in the Adv-Dis-I-Averse condition exhibited strong preferences in response for both moderate Dis-I and moderate Adv-I offers (30:70 and 70:30 splits, respectively).

### Contagion effects for extreme unfair offers suggest generalization

In Experiment 2, we took the same analysis approach as in Experiment 1, examining changes in rejection rates between the Baseline Phase and the Transfer Phase (after participants experienced feedback with moderately unfair offers). Similar to what we observed in Experiment 1 (*Figure 4a*), compared to the participants in the Dis-I-Averse Condition, participants in the Adv-I-Averse Condition increased their rates of rejection of extreme Adv-I (10:90) offers in the Transfer Phase, relative to the Baseline phase ($\beta$(*SE*)=0.12(0.04), p=0.004; *Supplementary file 1I*), suggesting that participants' learned (and adopted) Adv-I-averse preferences generalized from one specific offer type (30:70) to an offer type for which they received no Teacher feedback (10:90). Examining extreme Dis-I offers where the Teacher exhibited identical preferences across the two learning conditions, we found

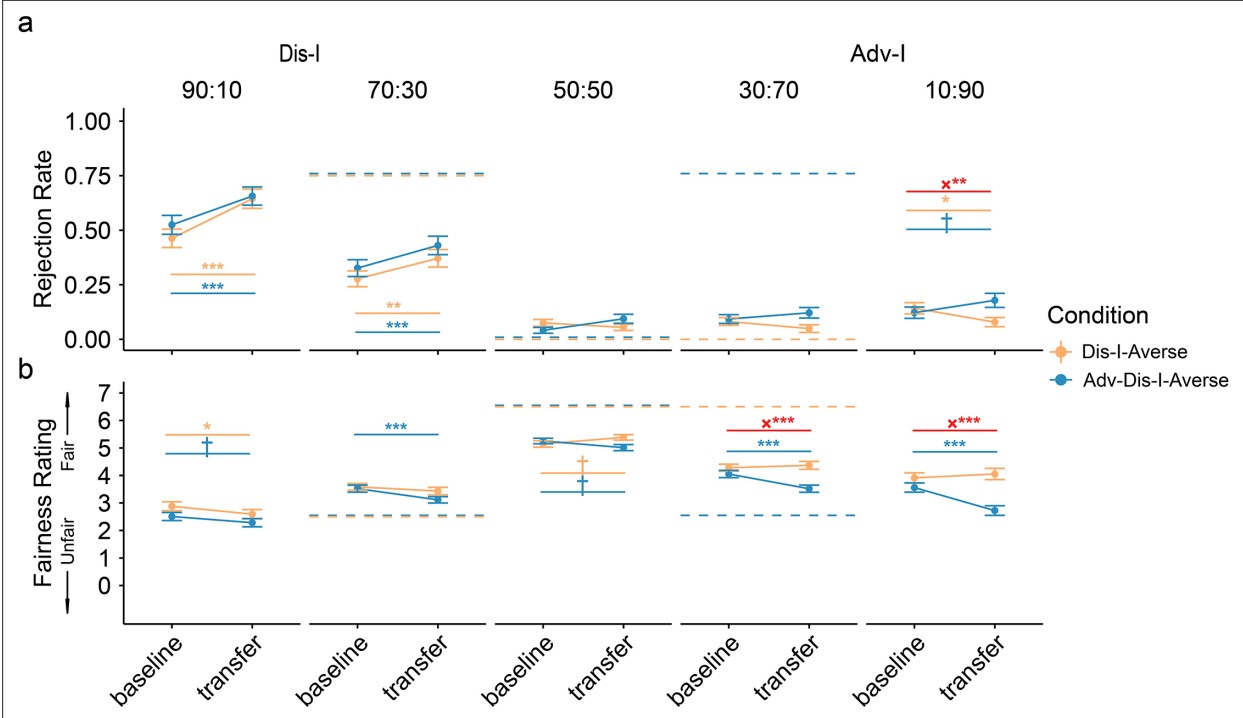

**Figure 4.** Baseline and transfer phase behavior in Experiment 2. (**a**) Contagion in extremely unfair offers. Although no feedback was provided in the Learning phase for 90:10 or 10:90 splits, we observed generalization of punishment preferences to these offers. Dashed lines represent the Teacher's preferences. (**b**) Fairness rating changes. We found significant changes from Baseline to Transfer phase in fairness rating for 90:10 in both Adv-Dis-I-Averse and Dis-I-Averse Condition, but only in Adv-Dis-I-Averse Condition for 10:90 offers. Dashed lines represent the Teacher's observed preferences (Panel a) and fairness ratings (Panel b). Error bars represent standard errors (†indicates p<0.1, *indicates p<0.05, **indicates p<0.01, ***indicates p<0.001). Resutls from linear mixed models (LMM, n = 97 for both conditions).

no difference in the Changes of Rejection Rates from Baseline to Transfer phase between conditions ($\beta(SE)$=-0.05(0.04). p<0.259). Mirroring the observed rejection rates (*Figure 4b*), relative to the Dis-I-Averse Condition, participants' fairness ratings for extreme Adv-I offers increased more from the Baseline to Transfer phase in the Adv-Dis-I-Averse Condition than in the Dis-I-Averse condition ($\beta(SE)$=-0.97(0.18), p<0.001), but, importantly, changes in fairness ratings for extreme Dis-I offers did not differ significantly between learning conditions ($\beta(SE)$=0.06(0.18), p<0.723).

In short, we found evidence that participants generalized across learning contexts, which in turn shaped their own rejection responses to extreme offers. In other words, it appears that preferences acquired through contagion extend beyond mere associations between single offers and actions and instead rely on a mechanism that infers the latent structure of the Teacher's fairness preferences.

## Preference changes in the learning phase suggest generalization

A primary goal of Experiment 2 was to demonstrate that learning the Teacher's preferences with respect to moderately unfair offers generalized to extremely unfair offers, for which no feedback from the Teacher was provided. The time course of rejection rates in Adv-I contexts during the Learning phase (*Figure 5*) reveals that participants learned over time to punish mildly unfair (30:70) offers, and these punishment preferences generalized to more extreme (10:90) offers. Specifically, compared to participants in the Dis-I-Averse Condition, participants in the Adv-Dis-I-Averse condition exhibited a significantly larger increase in rejection rates for 10:90 (Adv-I) offers (*Figure 5*, $\beta(SE)$=0.81(0.26), p=0.002; mixed-effects logistic regression, see *Supplementary file 1J*). Again, when comparing the rejection rate increase in the extremely Dis-I offers (90:10), we didn't find a significant difference between conditions ($\beta(SE)$=-0.25(0.19), p=0.707).

Finally, following Experiment 1, we fit a series of computational models of Learning phase choice behavior, comparing the goodness of fit of the five best-fitting models from Experiment 1 (see Materials and methods). As before, we found that the Preference Inference model provided the best fit

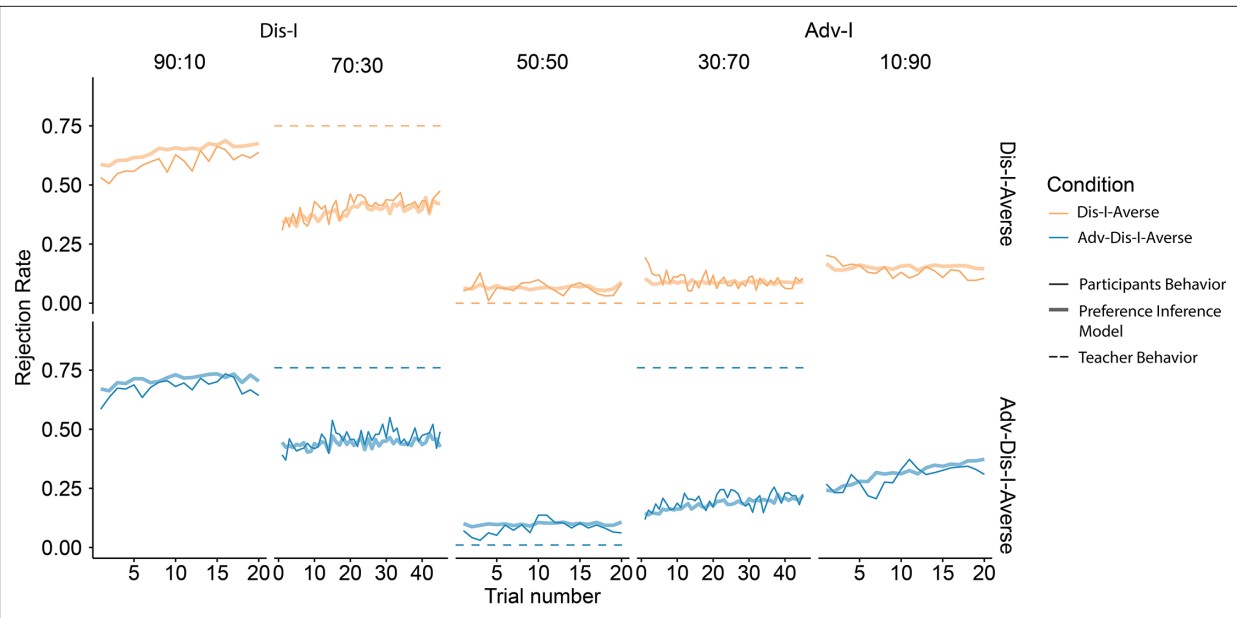

**Figure 5.** Learning phase choice behavior in Experiment 2. Learning effects were documented in extremely unfair offers. Rejection choices were summarized across subjects. Dashed lines indicate the Rejection choice of the Teacher. The learning effect was evident for 90:10 offers in Dis-I-Averse condition and 10:90 offers in Adv-Dis-I Averse condition. Thin solid lines represent participants' rejection choice, thick solid lines show the predictions of the Preference Inference Model, and the dashed lines indicate the Teacher's preferences (not observed by participants in 90:10 and 10:90 splits).

The online version of this article includes the following figure supplement(s) for figure 5:

**Figure supplement 1.** Model comparison results in Experiment 2.

of participants' Learning Phase behavior (*Figure 5—figure supplement 1*, *Supplementary file 1L*). Given that this model is able to infer the Teacher's underlying inequity-averse preferences (rather than learns offer-specific rejection preferences), it is unsurprising that this model best describes the generalization behavior observed in Experiment 2. We also simulated this model's Learning Phase behavior using participants' estimated parameter values and found that the model, mirroring participants' choices, exhibits clear incremental changes in rejection rates (*Figure 5*; thick lines), both for offers where the model received explicit feedback (70:30) and for offers where the model received no feedback (90:10). In other words, like participants, the model generalizes the learned inequity-averse preferences to extreme Adv-I offers (90:10), which stems from the model's trial-by-trial updating of the parameters governing the Teacher's preferences (see *Figure 5—figure supplement 1b, c*).

## Discussion

While people tend to reject proposed resource allocations where they stand to receive less than their peers (so-called disadvantageous inequity; Dis-I), they are markedly less averse to resource allocations where they stand to unfairly gain more than their peers (advantageous inequity or Adv-I; *Blake et al., 2015*; *Luo et al., 2018*). Here we considered the possibility that these complex, other-regarding preferences for fairness can be imparted merely by observing (and enacting) the preferences of another person. We investigated, in an Ultimatum Game setting, whether Adv-I-averse preferences can be shaped by learning and implementing the preferences of another individual. We leveraged a well-characterized observational learning paradigm (*FeldmanHall et al., 2018*), exposing participants to another individual (the Teacher) exhibiting a strong preference for punishment of advantageously unfair offers, and probed whether these punishment preferences in turn transferred to participants making choices on their own. We found that participants' own Adv-I-averse preferences shifted towards the preferences of the Teacher they just observed.

Previous work has outlined a number of important differences with respect to how individuals respond to advantageous (Adv-I) and disadvantageous inequity (Dis-I). Aversion to Dis-I is thought to arise from negative emotions such as spite (*McAuliffe et al., 2014*; *Pillutla and Murnighan, 1996*)

engendered by consideration of one's standing relative to others, while Adv-I aversion, in contrast, is thought to stem from concerns about fairness or inequality (*McAuliffe et al., 2013*). Hence, the expression of Adv-I aversion may signal, and even enforce, egalitarian social norms. Developmental evidence supports this distinction between Adv-I versus Dis-I aversion. While Dis-I aversion emerges at the age of 4, Adv-I aversion does not manifest until about 8 years (*Blake et al., 2015*; *Blake and McAuliffe, 2011*; *McAuliffe et al., 2017*). In fact, Adv-I aversion—which entails trading off self-interest against a social norm enforcement—is not even commonly observed in adults (*Blake et al., 2015*; *Hennig-Schmidt et al., 2008*; *Luo et al., 2018*), suggesting that Adv-I aversion is more difficult (and less likely) to be learned than Dis-I aversion. The observation that Adv-I averse preferences can be learned suggests that observational learning processes are a potent and promising means by which sophisticated fairness preferences can be imparted. However, it is important to note that our conclusions are based on the comparison between the Dis-I averse and Adv-Dis-I averse conditions, both of which expose participants to Teachers who exhibit some form of inequity-averse preferences. It is plausible that the Dis-I Averse condition cannot provide a 'pure' assessment of participants' default tendency to punish advantageous inequity, but rather participants in that condition may be learning—in contrast to participants in the Adv-Dis-I averse condition—to avoid rejecting Adv-I offers. However, we note that our previous work examining transmission of Dis-I-averse preferences (in a structurally similar paradigm) employed a control condition defined by a teacher exhibiting indifferent (random) preferences and found little evidence for spontaneous changes in punishment rates, across unfairness levels *FeldmanHall et al., 2018*. Still, work examining transmission of advantageous inequity aversion would benefit from stricter control conditions—either with no feedback or an indifferent teacher—to further clarify whether the rejection rates difference between these two conditions is driven by the Dis-I aversion or the Adv-Dis-I aversion condition.

Mechanistically, we found that participants' feedback-based learning of punishment preferences was best characterized by a computational model that assumes individuals infer the Teacher's latent and structured preferences for punishment—rather than a simple RL account assuming that individuals learn contextually bound punishment preferences. To support the interpretation that individuals indeed 'model' the fairness preferences of others, in a second experiment, we directly test whether participants can infer the Teachers' inequity-averse preferences across contexts. We found that participants generalized the Teacher's punitive preferences to other contexts that varied in their unfairness, and this occurred in both Adv-I and Dis-I contexts, suggesting that the discovery of latent structure is instrumental for generalization.

The representation of others' beliefs in other interpersonal decision-making tasks has been previously formalized, computationally, by a Bayesian account of theory of mind (ToM) in which the hypothetical beliefs were described by a prior distribution, and participants update this distribution using Bayesian updating (*Baker et al., 2009*; *Jara-Ettinger, 2019*). This computational framework has been applied to describing behaviors in a group decision-making task (*Khalvati et al., 2019*). In our Preference-inference model, on each trial, the learner makes a guess about the Teacher's inequity aversion parameters, then the Teacher's feedback is subsequently used to further constrain the range of parameter values which could conceivably produce the Teacher's observed feedback. The learner then updates their initial guess range. Conceptually, this learning mechanism is consistent with the sort of Bayesian updating.

One open question concerning the observed contagion effects is how the identities—and number—of teachers experienced in the Learning phase bear on the strength of the learning and contagion effects observed. For simplicity, our Learning Phase employed only one distinct Teacher, with no meaningful identity or social attributes, which contrasts with many social interactions in daily life, which are almost always accompanied by identifying information or attributes concerning the other (*Hester and Gray, 2020*). In contrast, our interactions with others are profoundly influenced by the identities of others—for example, whether they are conservative versus liberal (*Leong et al., 2020*), whether they are in- versus out-group members (*Hein et al., 2010*; *Vives et al., 2022*). At the same time, the strength of social influence often increases with the number (or proportion) of individuals in a group expressing a particular preference (*Cialdini and Goldstein, 2004*; *Son et al., 2019*). However, it may also be the case that social contagion effects require repeated interactions with the same individual (*Tsvetkova and Macy, 2014*), which the contagion observed in the present paradigm corroborates. Accordingly, future work should aim to examine the influence of the teacher's identity—and

its concordance with the learner's identity—as well as seek to understand how the relative balance of repeated experience with identical teachers versus the number of distinct teachers modulates the strength of contagion effects in Adv-I/Dis-I punishment preferences.

In summary, our study provides an initial demonstration that despite the desire for self-gain, we observe that people can swiftly and readily acquire another's preferences for advantageous inequity, even when it comes at a monetary cost to the self. Computationally, we find that this contagion of inequity-averse preferences occurs through representing the underlying structure of another's preferences, rather than a RL-like process of learning simple context-action associations. Importantly, these inferred preferences are sufficient to induce individuals to change their preferences for punishing advantageous inequity, suggesting that social influence may be one promising route through which social norm enforcement—uncommonly observed in the case of Adv-I—can be promoted.

## Materials and methods

### Participants

We recruited US-based participants from Amazon Mechanical Turk (*Crump et al., 2013*) for both Experiment 1 (N=200, *M* age = 37.53 (SD = 10.88), 75 females) and Experiment 2 (N=200, *M* age = 37.16 (SD = 11.79) years old, 80 females). These sample sizes are based on our previous work employing the same per-condition sample sizes (*FeldmanHall et al., 2018*) in a punishment-learning task. Participants provided informed consent in accordance with the McGill University Research Ethics Board (#258–1118). Participants were randomly assigned to either the Adv-Dis-I-Averse (N=100) or the Dis-I-Averse (N=100) condition. As we replicated all key reported results when excluding participants who evidenced some disbelief that the Teacher (see below for procedural details), suggesting that these results are robust to potential disbelief about the task structure as described to them. In Experiment 2, we excluded the data of participants who failed to meet the requirements of each analysis due to missed trials (3 in Adv-Dis-I-Averse and 3 in Dis-I-Averse Conditions were removed in Contagion effects analysis).

### Rejection learning paradigm

We used a modified version of the Ultimatum Game (*Güth et al., 1982*) to probe participants' fairness preference. In this task, the proposer offers an allocation of a total amount (e.g. $1 out of $10). Then, another player, the Receiver, chooses to reject or accept the proposed allocation. If the Receiver accepts, both players receive the proposed amount; if they reject, both players receive nothing. Following our past work examining contagion effects (*FeldmanHall et al., 2018*), our task consisted of three phases: Baseline, Learning, and Transfer (see *Figure 1a*).

In the Baseline and Transfer phases, participants responded to unfair offers as a receiver in multiple rounds of ultimatum game, and participants were informed that the Proposer on each trial was a different, randomly chosen individual with a unique (fictitious) participant ID (*Figure 1c*), which permitted us to measure participants' preferences and beliefs about fairness irrespective of any particular Proposer. We considered 5 unique offer levels (90:10, 70:30, 50:50, 30:70, 10:90) ranging from extreme Dis-I to extreme Adv-I, including 'fair' offers (50:50), which allowed us to measure participants' baseline rejection tendency. We instructed participants that these Proposers were concurrently participating in this task, and these were 'real time' offers. Critically, participants were responding to offers made by fictitious players with predetermined offers to ensure that we could observe rejection preferences for each offer level. On 3/5 of Baseline and Transfer Phase trials, participants also rated the fairness of the offer on a 1–7 scale (1 being 'Strongly unfair'; 7 being 'Strongly fair') after making an accept/reject choice. Participants experienced five trials of each offer type in both the Baseline and Transfer Phase.

In the Learning phase, participants played an Ultimatum Game, acting as a third party deciding to accept or reject Proposers' offers for another receiver (the Teacher) receiving offers from Proposers (see *Figure 1d*). After the participant chose to accept or reject the offer, the Teacher's preferred response was revealed. In the Adv-Dis-I-Averse condition, the Teacher indicated preference for rejection of fair (50:50) and Adv-I offers and uniformly rated these Adv-I offers as unfair (see *Figure 1b*). In the Dis-I-Averse condition, the Teacher accepted all fair and Adv-I offers and rated these offers as uniformly fair. In both the Dis-I-Averse and Adv-Dis-I-Averse, Dis-I offers were rejected and rated as

unfair. The Teachers' rejection rates and ratings in response to each offer type are provided in *Supplementary file 1A and B*.

Again, on 3/5 of the trials, participants rated the fairness of each offer, and on these trials, participants also saw the Teachers' fairness rating of the offer. Like the Baseline and Transfer phases, participants saw a unique, randomly chosen Proposer on every trial but were informed that there was a single, unvarying Teacher over the entire Learning Phase. In Experiment 1, participants experienced 20 trials for each of the five offer types considered (90:10, 70:30, 50:50, 30:70, 10:90).

In all task phases, we added uniformly distributed noise to each trial's offer (ranging from –9 to 9, inclusive, rounding to the nearest integer) such that the random amount added (or subtracted) from the Proposer's share was subtracted (or added) to the Receiver's share. We adopted this manipulation to make the proposers' behavior appear more realistic. The orders of offers participants experienced were fully randomized within each experiment phase.

Experiment 2 followed the same procedure as Experiment 1 except for the following changes. First, and most importantly, in the Learning Phase, participants did not experience feedback on extreme Adv-I offers (10:90 splits). Second, we added 25 additional trials of each offer type (70:30 and 30:70 splits) to the Learning Phase to provide increased opportunity to observe the Teacher's preferences. Third, participants were instructed that Proposers' offers (which were predetermined) were generated by previous participants in previous similar experiments. Finally, in all task phases, participants were required to respond within a 3 s deadline when making choices, and a 4 s deadline when providing fairness ratings.

## Data analysis

We used mixed-effects regression, implemented in the 'lmer' package for R (*Bates et al., 2015*) to estimate the effect of offer level and punishment condition upon contagion rates— the rejection rate change between the Baseline and Transfer phases. To do this, the rejection rate change was modeled by interactions between Offer types (factors of five levels: 90:10, 70:30, 50:50, 30:70, 10:90, coded using 5 dummy-coded columns) and Condition (Adv-Dis-I-Averse vs. Dis-I-Averse, coded using 2 dummy-coded columns), with random intercepts taken over participants (see *Supplementary file 1G* for full coefficient estimates).

We estimated changes in rejection rates in the Learning Phase using mixed-effects logistic regressions in Learning. Specifically, rejection choices were predicted as a function of trial number, offer type, condition, and their resultant interactions, taking the two-way interactions between Trial number and offer type (random slope) as random effects over participants. To estimate changes in fairness ratings, we estimated linear mixed-effects models with the same terms.

## Computational models of learning phase behavior

We considered seven computational models of Learning Phase choice behavior, which we fit to individual participants' observed sequences of choices, in Experiment 1, via Maximum Likelihood Estimation. Importantly, Models 1 (Random choosing) and 2 (Static preference) are baseline models which assume that rejection probabilities are fixed over time, while all other models allow for learning of rejection rates over time in accordance with Teacher feedback.

### Model 1 (random choosing)

This model assumes that participants reject offers with a fixed probability governed by the parameter $p_{\text{offertype}}$ (one for each offer type; 5 free parameters in total).

$$P_{reject}\left(offertype\right) \sim Bernoulli\left(p_{offertype}\right) \tag{1}$$

### Model 2 (static preference)

This model assumes that when choosing for others, the utility of accepting an offer, relative to rejection of the offer, is governed by the Fehr-Schmidt (FS) inequity aversion (*Fehr and Schmidt, 1999*; *Luo et al., 2018*) function:

$$U_{accept}\left(offer\right) = \text{offer} - \alpha * max\left(50 - offer, 0\right) - \beta * max\left(offer - 50, 0\right) \tag{2}$$

$$U_{reject}\left(offer\right) = 0 \tag{3}$$

In this function, *offer* represents the share the Proposer gives to the Receiver, $\alpha$ parameterizes the Teacher's disutility (or 'envy') in accepting disadvantageous unfair offers (Dis-I aversion), and $\beta$ captures Teacher's disutility (or 'guilt') for accepting advantageous unfair offers (Adv-I aversion), which can each range from 0 to 10.

These action utilities were then transformed to choice probabilities using the softmax choice rule:

$$P_{reject}\left(offer\right) = exp\left(\tau * U_{reject}\right) / \left(exp\left(\tau * U_{reject}\right) + exp\left(\tau * U_{accept}\right)\right) \tag{4}$$

where the inverse temperature ($\tau$) parameter captures decision noise, such that a larger $\tau$ corresponds to a higher probability of choosing the action with larger utility, and as $\tau$ approaches 0, the two options are chosen with equal probability. In total, this model has three free parameters.

## Model 3 (basic RL)

Model 3 is a simple RL model following that used by **FeldmanHall et al., 2018** only one learning rate, which represents and updates values of the two actions separately for each offer type, using a delta updating rule:

$$Q_{t+1}\left(action_t, offertype_t\right) = Q_t\left(action_t, offertype_t\right) + \eta * \left(R_t - Q_t\left(action_t, offertype_t\right)\right) \tag{5}$$

where $action_t$ is the action the participant chose (accept or reject) on the $t$-th trial, $R_t$ is the reward on the $t$-th trial, which is defined as 1 (reward obtained) when the action taken was the same as the action the Teacher would have preferred, and 0 (no reward) otherwise. The softmax choice rule was used to translate these action values to predicted choice probabilities. We kept the initial values fixed in this model, that is $Q_0(reject, offertype)$= 0.5, ($offertype \in$ 90:10, 70:30, 50:50, 30:70, 10:90). This model has two free parameters.

## Model 4 (offer-sensitive RL)

This RL model is a more complex variant of Model 3 and assumes a separate learning rate for each offer type. Similar to Model 3, we set the initial Q values to 0.5 for each offer type. In total, this model has six free parameters.

## Model 5 (offer-sensitive RL with separate initial values)

This RL model extends Model 3, assuming different initial action values for each offer type. Formally, this model treats $Q_0(reject, offertype)$, ($offertype \in$ 90:10, 70:30, 50:50, 30:70, 10:90) as free parameters with values between 0 and 1, resulting in seven free parameters.

## Model 6 (preference inference)

Model 6 posits that the participant infers the Fehr-Schmidt utility function (**Equation 2**) governing the Teacher's preferences, and updates their modeled $\alpha$ and $\beta$ parameters incrementally from feedback, under the assumption that the Teacher's indicated choices are made in accordance with each offer's Fehr-Schmidt utility (more formally, the Teacher rejects the offer when $U_{accept}\left(offer\right) < 0$). As $\alpha$ and $\beta$ govern the disutility of unfair offers, the model infers the minimal value of $\alpha$ (or $\beta$) that would lead to rejection of Dis-I (or Adv-I) offers, and similarly, the maximum values of $\alpha$ (or $\beta$) that would lead to acceptance of Dis-I (or Adv-I) offers.

Accordingly, after observing that the Teacher prefers rejection in response to a Dis-I offer, **Equation 2** can be transformed to the following inequality:

$$U_{accept}\left(offer\right) = offer - \alpha * \left(50 - offer\right) < 0 \tag{6}$$

where the model can infer a lower bound of $\alpha$, that would lead to the offer's rejection by solving (6):

$$\alpha > \frac{offer}{50 - offer} \tag{7}$$

The right side of (7) can be denoted as $\alpha_{lb}$, and then that trial's estimate of $\alpha$ (denoted $\alpha_t$) is updated as follows:

$$\alpha_{t+1} = \begin{cases} \alpha_t, \, if \, \alpha_t > \alpha_{lb} \\ \alpha_t + \eta * (\alpha_{lb} - \alpha_t), otherwise \end{cases} \tag{8}$$

The parameter $\eta$ governs the rate at which the learner's estimate of the Teacher's $\alpha$ value is updated and is constrained to the range [0, 5].

The updating procedure is similar when the Teacher indicates acceptance of a Dis-I offer, which implies that $U_{accept}(offer) > 0$, and the following inequality:

$$U_{accept}(offer) = offer - \alpha * (50 - offer) > 0 \tag{9}$$

which yields an upper bound for the envy parameter $\alpha$:

$$\alpha < \frac{offer}{50 - offer} \tag{10}$$

This upper bound, $\alpha_{ub}$, is in turn used to update $\alpha_t$:

$$\alpha_{t+1} = \begin{cases} \alpha_t, \, if \, \alpha_t < \alpha_{lb} \\ \alpha_t + \eta * (\alpha_{ub} - \alpha_t), \, otherwise \end{cases} \tag{11}$$

In the case of Adv-I offers, the model employs the identical procedure to update the 'guilt' parameter $\beta$, and $\alpha$ is updated only in Dis-I offers, while $\beta$ is only updated in Adv-I offers. Following **Luo et al., 2018**, $\alpha$ and $\beta$ were restricted to the range of [0, 10]. The initial value of $\alpha$ and $\beta$ is taken as free parameters in the range of [0, 10], resulting in a model with a total of 4 free parameters. This preference inference model predicts the generalization effects, as experience of the teacher's rejection in moderately unfair offers already allows the updating of both Fehr-Schmidt parameters ($\alpha$ and $\beta$).

## Model 7 (similarity RL)

Model 7 assumes that while participants learn the value for each action by trial and error (as in Models 1–5), they can generalize learning across offers based on the similarity of the current (experienced) offer type to other (non-experienced) offer types. This generalization is governed by a normal distribution whereby generalization decreases with increasing similarity. Formally, this model assumes participants track $Q$ values corresponding to every possible offer—that is, Q(reject, *offer*) and Q(accept, *offer*) for all integer offers from 1 to 100. After experiencing Teacher feedback for an experienced offer, the model then updates the action values for all offers:

$$Q_{t+1}(action_t, offer) = Q_t(action_t, offer) + \eta_{offer} * (R_t - Q_t(action_t, offer)) \tag{12}$$

Here, $\eta_{\text{offer}}$ is a learning rate computed by scaling the learning rate parameter $\eta$ with respect to the difference between the just-experienced offer on trial t and the to-be-updated offer:

$$\eta_{offer} = \eta * f(offer - offer_t) / f(0) \tag{13}$$

such that

$$f(x) = \frac{1}{\sigma\sqrt{2\pi}} \exp\left(\frac{-x^2}{2\sigma^2}\right) \tag{14}$$

where $\sigma$ is a free parameter governing the width of the Gaussian, bound by the range [0,200]. To mimic the behavior of the Preference Inference model (Model 6), we modeled the starting value for rejection as a V-shape function. That is:

$$Q_0(reject, offer) = \max\left(\frac{\alpha}{50} * (50 - offer), \frac{\beta}{50} * (offer - 50)\right) \tag{15}$$

where $\alpha$ and $\beta$ are parameters (in the range [0,3]) governing the slope of the V-shape function.

## Model fitting and validation

All models were fit via Maximum Likelihood Estimation, employing a nonlinear optimization procedure using 100 random start points in the parameter space in order to find the best-fitting parameter values for each participant. We then computed the Akaike Information Criterion (AIC; *Akaike, 1974*) to select the best-fitting models of Learning phase choice behavior, penalizing each model's goodness-of-fit score by its complexity (i.e. number of free parameters). See *Supplementary file 1H, L* for parameter estimates and goodness-of-fit metrics. Note that in Experiment 2, we did not evaluate Models 3 and 4 because on account of their poor performance in Experiment 1.

Finally, to verify if the free parameters of the winning model (Preference Inference) are recoverable, we simulated 200 artificial subjects, based on the Learning Phase of Experiment 1, with free parameters randomly chosen (uniformly) from their defined ranges. We then employed the same model-fitting procedure as described above to estimate these parameter values. We found that all parameters of this model could be recovered (see *Figure 3—figure supplement 1*).

## Acknowledgements

This work was funded by the European Union (ERC Starting Grant, NEUROGROUP, 101041799) and by an NSERC Discovery Grant, a New Researchers Startup Grant from the Fonds de Recherche du Québec - Nature et Technologies, and an infrastructure award from the Canadian Foundation for Innovation. Views and opinions expressed are, however, those of the authors only and do not necessarily reflect those of the European Union or the European Research Council Executive Agency. Neither the European Union nor the granting authority can be held responsible for them.

## Additional information

### Funding

| Funder | Grant reference number | Author |
| --- | --- | --- |
| European Union | 101041799 | |
| Natural Sciences and Engineering Research Council of Canada | Discovery Grant | A Ross Otto |
| Fonds de Recherche du Québec – Nature et Technologies | New Researchers Startup Grant | A Ross Otto |
| Canada Foundation for Innovation | | A Ross Otto |

The funders had no role in study design, data collection and interpretation, or the decision to submit the work for publication.

### Author contributions

Shen Zhang, Conceptualization, Data curation, Formal analysis, Validation, Visualization, Methodology, Writing – original draft, Writing – review and editing; Oriel FeldmanHall, Conceptualization, Supervision, Methodology, Writing – review and editing; Sébastien Hétu, Conceptualization, Supervision, Investigation, Methodology, Writing – review and editing; A Ross Otto, Conceptualization, Supervision, Funding acquisition, Investigation, Methodology, Project administration, Writing – review and editing

### Author ORCIDs

Shen Zhang ⓘ https://orcid.org/0000-0002-5801-286X
Oriel FeldmanHall ⓘ https://orcid.org/0000-0002-0726-3861
Sébastien Hétu ⓘ http://orcid.org/0000-0002-5323-931X
A Ross Otto ⓘ https://orcid.org/0000-0002-9997-1901

## Ethics

Human subjects: Participants provided informed consent in accordance with the McGill University Research Ethics Board. The ethical approval number is 258-1118.

Reviewer #1 (Public review): https://doi.org/10.7554/eLife.102800.3.sa1
Reviewer #2 (Public review): https://doi.org/10.7554/eLife.102800.3.sa2
Author response https://doi.org/10.7554/eLife.102800.3.sa3

---

# Additional files

## Supplementary files

Supplementary file 1. Model reports.

MDAR checklist

## Data availability

All behavioral data and analysis code have been deposited at https://osf.io/6xn5b and are publicly available.

The following dataset was generated:

| Author(s) | Year | Dataset title | Dataset URL | Database and Identifier |
| --- | --- | --- | --- | --- |
| Zhang S, Otto AR | 2025 | Fairness Contagion | https://doi.org/10. 17605/OSF.IO/6XN5B | Open Science Framework, 10.17605/OSF.IO/6XN5B |

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
