## [Editor Report · eLife Assessment]

This cleverly designed and potentially **important** work supports our understanding regarding how and whether social behaviours promoting egalitarianism can be learned, even when implementing these norms entails a cost for oneself. However, the evidence supporting the major claims is currently **incomplete**, with the major limitation being whether Ps truly learn egalitarianism from a teacher or instead exhibit reduced guilt across time that is reduced when observing others behaving more selfishly. With a strengthening of the supporting evidence, this work will be of interest to a wide range of fields, including cognitive psychology/neuroscience, neuroeconomics, and social psychology, as well as policy making.

---

## [Referee Report · Reviewer #1 (Public review)]

Summary:

Zhang et al. addressed the question of whether advantageous and disadvantageous inequality aversion can be vicariously learned and generalized. Using an adapted version of the ultimatum game (UG), in three phases, participants first gave their own preference (baseline phase), then interacted with a "teacher" to learn their preference (learning phase), and finally were tested again on their own (transfer phase). The key measure is whether participants exhibited similar choice preference (i.e., rejection rate and fairness rating) influenced by the learning phase, by contrasting their transfer phase and baseline phase. Through a series of statistical modeling and computational modeling, the authors reported that both advantageous and disadvantageous inequality aversion can indeed be learned (Study 1), and even be generalised (Study 2).

Strengths:

This study is very interesting, that directly adapted the lab's previous work on the observational learning effect on disadvantageous inequality aversion, to test both advantageous and disadvantageous inequality aversion in the current study. Social transmission of action, emotion, and attitude have started to be looked at recently, hence this research is timely. The use of computational modeling is mostly appropriate and motivated. Study 2 that examined the vicarious inequality aversion on conditions where feedback was never provided is interesting and important to strengthen the reported effects. Both studies have proper justifications to determine the sample size.

Weaknesses:

Despite the strengths, a few conceptual aspects and analytical decisions have to be explained, justified, or clarified.

INTRODUCTION/CONCEPTUALIZATION

(1) Two terms seem to be interchangeable, which should not, in this work: vicarious/observational learning vs preference learning. For vicarious learning, individuals observe others' actions (and optionally also the corresponding consequence resulted directly by their own actions), whereas, for preference learning, individuals predict, or act on behalf of, the others' actions, and then receive feedback if that prediction is correct or not. For the current work, it seems that the experiment is more about preference learning and prediction, and less so about vicarious learning. But the intro and set are heavily around vicarious learning, and late the use of vicarious learning and preference learning is rather mixed in the text. I think either tone down the focus on vicarious learning, or discuss how they are different. Some of the references here may be helpful: Charpentier et al., Neuron, 2020; Olsson et al., Nature Reviews Neuroscience, 2020; Zhang & Glascher, Science Advances, 2020

EXPERIMENTAL DESIGN

(2) For each offer type, the experiment "added a uniformly distributed noise in the range of (-10 ,10)". I wonder how this looks like? With only integers such as 25:75, or even with decimal points? More importantly, is it possible to have either 70:30 or 90:10 option, after adding the noise, to have generated an 80:20 split shown to the participants? If so, for the analyses later, when participants saw the 80:20 split, which condition did this trial belong to? 70:30 or 90:10? And is such noise added only to the learning phase, or also to the baseline/transfer phases? This requires some clarification.

(3) For the offer conditions (90:10, 70:30, 50:50, 30:70, 10:90) - are they randomized? If so, how is it done? Is it randomized within each participants, and/or also across participants (such that each participant experienced different trial sequences)? This is important, as the order especially for the leanring phase can largely impact on the preference learning of the participants.

STATISTICAL ANALYSIS & COMPUTATIONAL MODELING

(4) In Study 1 DI offer types (90:10, 70:30), the rejection rate for DI-AI averse looks consistently higher than that for DI averse (ie, blue line is above the yellow line). Is this significant? If so, how come? Since this is a between-subject design, I would not anticipate such a result (especially for the baseline). Also, for the LME results (eg, Table S3), only interactions were reported but not the main results.

(5) I do not particularly find this analysis appealing: "we examined whether participants' changes in rejection rates between Transfer and Baseline, could be explained by the degree to which they vicariously learned, defined as the change in punishment rates between the first and last 5 trials of the Learning phase." Naturally, participants' behavior in the first 5 trials in the learning phase will be similar to those in the baseline; and their behavior in the last 5 trials in the learning phase would echo those at the transfer phase. I think it would be stronger to link the preference learning results to the chance between baseline and transfer phase, eg, by looking at the difference between alpha (beta) at the end of the learning phase and the initial alpha (beta).

(6) I wonder if data from the baseline and transfer phases can also be modeled, using a simple Fehr-Schimdt model? This way, the change in alpha/beta can also be examined between the baseline and transfer phase.

(7) I quite liked Study 2 that tests the generalization effect, and I expected to see an adapted computational modeling to directly reflect this idea. Indeed, the authors wrote "[...] given that this model [...] assumes the sort of generalization of preferences between offer types [...]". But where exactly did the preference learning model assumed the generalization? In the methods, the modeling seems to be only about Study 1; did the authors advise their model to accommodate Study 2? The authors also ran simulation for the learning phase in Study 2 (Figure 6), and how did the preference updated (if at all) for offers (90:10 and 10:90) where feedback was not given? Extending/Unpacking the computational modeling results for Study2 will be very helpful for the paper.

Comments on revisions:

I kept my original public review, so that future readers can see the progress and development of the manuscript.

The authors have largely addressed my original questions/concerns, and I have two outstanding comments.

(a) Related to my original comment #6, where I suggested to apply the F-S model also to the baseline and transfer phase. The authors were inclined not to do it, but in fact later in comment #7 and in the manuscript they opted to use a more complex F-S-based model to their learning phase. I agree that the rejection rate is indeed a clear indication, but for completeness, it'd be more consistent and compelling if the paper follows a model-free (model-agnostic) and model-based approach in all phases of the experiment.

(b) Related to my original comment #4, I appreciate that the authors have provided more details of their LMM models. But I don't think it is accurate regardless.First, all offer levels (50:50, 30:70, 10:90), should not be coded as pure categorical levels. In fact, they have an ordinal meaning, a single ordinal predictor with three levels should be used. This also avoids the excessive number of interactions the authors have pointed out.

Second, running a model with only interactions without main effects is flawed. All textbooks on stats emphasize that without the presence of the main effects, the interpretation of interaction only is biased.

So these LMMs needs to be revised before the manuscript eventually gets to a version of record.

---

## [Referee Report · Reviewer #2 (Public review)]

Summary:

This study investigates whether individuals can learn to adopt egalitarian norms that incur a personal monetary cost, such as rejecting offers that benefit them more than the giver (advantageous inequitable offers). While these behaviors are uncommon, two experiments aim to demonstrate that individuals can learn to reject such offers by observing a "teacher" who follows these norms. The authors use computational modelling to argue that learners adopt these norms through a sophisticated process, inferring the latent structure of the teacher's preferences, akin to theory of mind.

Strengths:

This paper is well-written and tackles an important topic relevant to social norms, morality, and justice. The findings are promising (though further control conditions are necessary to support the conclusions). The study is well-situated in the literature, with a clever experimental design and a computational approach that may offer insights into latent cognitive processes. In the revision, the authors clarified some questions related to the initial submission.

Weaknesses:

Despite these strengths, I remain unconvinced that the current evidence supports the paper's central claims. Below, I outline several issues that, in my view, limit the strength of the conclusions.

(1) Experimental Design and Missing Control Condition:

The authors set out to test whether observing a "teacher" who is averse to advantageous inequity (Adv-I) will affect observers' own rejection of Adv-I offers. However, I think the design of the task lacks an important control condition needed to address this question. At present, participants are assigned to one of two teachers: DIS or DIS+ADV. Behavioral differences between these groups can only reveal relative differences in influence; they cannot establish whether (and how) either teacher independently affects participants' own behavior. For example, a significant difference between conditions can emerge even if participants are only affected by the DIS teacher and are not affected at all by the DIS+ADV teacher. What is crucially missing here is a no-teacher control condition, which can then be compared with each teacher condition separately. This control condition would also control for pure temporal effects unrelated to teacher influence (e.g., increasing Adv-I rejections due to guilt build-up).

While this criticism applies to both experiments, it is especially apparent in Experiment 2. As shown in Figure 4, the interaction for 10:90 offers reflects a decrease in rejection rates following the DIS teacher, with no significant change following the DIS+ADV teacher. Ignoring temporal effects, this pattern suggests that participants may be learning NOT to reject from the DIS teacher, rather than learning to reject from the DIS+ADV teacher. On this basis, I do not see convincing evidence that participants' own choices were shaped by observing Adv-I rejections.

In the Discussion, the authors write that "We found that participants' own Adv-I-averse preferences shifted towards the preferences of the Teacher they just observed, and the strength of these contagion effects related to the degree of behavior change participants exhibited on behalf of the Teachers, suggesting that they internalized, at least somewhat, these inequity preferences." However, there is no evidence that directly links the degree of behaviour change (on the teacher's behalf) to contagion effects (own behavioural change). I think there was a relevant analysis in the original version, but it was removed from the current version.

(2) Modelling Efforts:The modelling approach is underdeveloped. The identification of the "best model" lacks transparency, as no model-recovery results are provided. Additionally, behavioural fits for the losing models are not shown, leaving readers in the dark about where these models fail. Readers would benefit from seeing qualitative/behavioural patterns that favour the winning model.Moreover, the reinforcement learning (RL) models used are overly simplistic, treating actions as independent when they are likely inversely related. For example, the feedback that the teacher would have rejected an offer provides evidence that rejection is "correct" but also that acceptance is "an error," and the latter is not incorporated into the modelling. In other words, offers are modelled as two-armed bandits (where separate values are learned for reject and accept actions), but the situation is effectively a one-armed bandit (if one action is correct, the other is mistaken). It is unclear to what extent this limitation affects the current RL formulations. Can the authors justify/explain their reasoning for including these specific variants? The manuscript only states Q-values for reject actions, but what are the Q-values for accept actions? This is unclear.

In Experiment 2, only the preferred model is capable of generalization, so it is perhaps unsurprising that this model "wins." However, this does not strongly support the proposed learning mechanism, lacking a comparison with simpler generalizing mechanisms (see following comments).

(3) Conceptual Leap in Modelling Interpretation:The distinction between simple RL models and preference-inference models seems to hinge on the ability to generalize learning from one offer to another. Whereas in the RL models, learning occurs independently for each offer (hence no cross-offer generalization), preference inference allows for generalization between different offers. However, the paper does not explore "model-free" RL models that allow generalization based on the similarity of features of the offers (e.g., payment for the receiver, payment for the offer-giver, who benefits more). Such models are more parsimonious and could explain the results without invoking a theory of mind or any modelling of the teacher. In such model versions, a learner acquires a functional form that allows prediction of the teacher's feedback based on offer features (e.g., linear or quadratic weighting). Because feedback for an offer modulates the parameters of this function (feature weights), generalization occurs without necessarily evoking any sophisticated model of the other person. This leaves open the possibility that RL models could perform just as well or even outperform the preference learning model, casting doubt on the authors' conclusions.

Of note: even the behaviourists knew that when Little Albert was taught to fear rats, this fear generalized to rabbits. This could occur simply because rabbits are somewhat similar to rats. But this doesn't mean Little Albert had a sophisticated model of animals that he used to infer how they behave.

In their rebuttal letter, the authors acknowledge these possibilities, but the manuscript still does not explore or address alternative mechanisms.

(4) Limitations of the Preference-Inference Model:The preference-inference model struggles to capture key aspects of the data, such as the increase in rejection rates for 70:30 DI offers during the learning phase (e.g., Fig. 3A, AI+DI blue group). This is puzzling.Thinking about this, I realized the model makes quite strong, unintuitive predictions which are not examined. For example, if a subject begins the learning phase rejecting the 70:30 offer more than 50% of the time (meaning the starting guilt parameter is higher than 1.5), then, over learning, the tendency to reject will decrease to below 50% (the guilt parameter will be pulled down below 1.5). This is despite the fact that the teacher rejects 75% of the offers. In other words, as learning continues, learners will diverge from the teacher. On the other hand, if a participant begins learning by tending to accept this offer (guilt < 1.5), then during learning, they can increase their rejection rate but never above 50%. Thus, one can never fully converge on the teacher. I think this relates to the model's failure in accounting for the pattern mentioned above. I wonder if individuals actually abide by these strict predictions. In any case, these issues raise questions about the validity of the model as a representation of how individuals learn to align with a teacher's preferences (given that the model doesn't really allow for such an alignment).

In their rebuttal letter, the authors acknowledged these anomalies and stated that they were able to build a better model (where anomalies are mitigated, though not fully eliminated). But they still report the current model and do not develop/discuss alternatives. A more principled model may be a Bayesian model where participants learn a belief distribution (rather than point estimates) regarding the teacher's parameters.

(5) Statistical Analysis:The authors state in their rebuttal letter that they used the most flexible random effect structure in mixed-effects models. But this seems not to be the case in the model reported in Table SI3 (the very same model was used for other analyses too). Indeed, here it seems only intercepts are random effects. This left me confused about which models were used.

---

## [Author Response]

The following is the authors’ response to the original reviews

**Reviewer #1 (Public review):**
Summary:Zhang et al. addressed the question of whether advantageous and disadvantageous inequality aversion can be vicariously learned and generalized. Using an adapted version of the ultimatum game (UG), in three phases, participants first gave their own preference (baseline phase), then interacted with a "teacher" to learn their preference (learning phase), and finally were tested again on their own (transfer phase). The key measure is whether participants exhibited similar choice preferences (i.e., rejection rate and fairness rating) influenced by the learning phase, by contrasting their transfer phase and baseline phase. Through a series of statistical modeling and computational modeling, the authors reported that both advantageous and disadvantageous inequality aversion can indeed be learned (Study 1), and even be generalised (Study 2).Strengths:This study is very interesting, it directly adapted the lab's previous work on the observational learning effect on disadvantageous inequality aversion, to test both advantageous and disadvantageous inequality aversion in the current study. Social transmission of action, emotion, and attitude have started to be looked at recently, hence this research is timely. The use of computational modeling is mostly appropriate and motivated. Study 2, which examined the vicarious inequality aversion in conditions where feedback was never provided, is interesting and important to strengthen the reported effects. Both studies have proper justifications to determine the sample size.Weaknesses:Despite the strengths, a few conceptual aspects and analytical decisions have to be explained, justified, or clarified.INTRODUCTION/CONCEPTUALIZATION(1) Two terms seem to be interchangeable, which should not, in this work: vicarious/observational learning vs preference learning. For vicarious learning, individuals observe others' actions (and optionally also the corresponding consequence resulting directly from their own actions), whereas, for preference learning, individuals predict, or act on behalf of, the others' actions, and then receive feedback if that prediction is correct or not. For the current work, it seems that the experiment is more about preference learning and prediction, and less so about vicarious learning. The intro and set are heavily around vicarious learning, and later the use of vicarious learning and preference learning is rather mixed in the text. I think either tone down the focus on vicarious learning, or discuss how they are different. Some of the references here may be helpful: (Charpentier et al., Neuron, 2020; Olsson et al., Nature Reviews Neuroscience, 2020; Zhang & Glascher, Science Advances, 2020)

We are appreciative of the Reviewer for raising this question and providing the reference. In response to this comment we have elected to avoid, in most cases, use of the term ‘vicarious’ and instead focus the paper on learning of others’ preferences (without specific commitment to various/observational learning per se). These changes are reflected throughout all sections of the revised manuscript, and in the revised title. We believe this simplified terminology has improved the clarity of our contribution.

EXPERIMENTAL DESIGN(2) For each offer type, the experiment "added a uniformly distributed noise in the range of (-10 ,10)". I wonder what this looks like? With only integers such as 25:75, or even with decimal points? More importantly, is it possible to have either 70:30 or 90:10 option, after adding the noise, to have generated an 80:20 split shown to the participants? If so, for the analyses later, when participants saw the 80:20 split, which condition did this trial belong to? 70:30 or 90:10? And is such noise added only to the learning phase, or also to the baseline/transfer phases? This requires some clarification.

We thank the Reviewer for pointing this out. The uniformly distributed noise was added to all three phases to make the proposers’ behavior more realistic. This added noise was rounded to integer numbers, constrained from -9 to 9, which means in both 70:30 and 90:10 offer types, an 80:20 split could not occur. We have made this feature of our design clear in the Method section Line 524 ~ 528:

“In all task phases, we added uniformly distributed noise to each trial’s offer (ranging from -9 to 9, inclusive, rounding to the nearest integer) such that the random amount added (or subtracted) from the Proposer’s share was subtracted (or added) to the Receiver’s share. We adopted this manipulation to make the proposers’ behavior appear more realistic. The orders of offers participants experienced were fully randomized within each experiment phase. ”

(3) For the offer conditions (90:10, 70:30, 50:50, 30:70, 10:90) - are they randomized? If so, how is it done? Is it randomized within each participant, and/or also across participants (such that each participant experienced different trial sequences)? This is important, as the order especially for the learning phase can largely impact the preference learning of the participants.

We agree with the Reviewer the order in which offers are experienced could be very important. The order of the conditions was randomized independently for each participant (i.e. each participant experienced different trial sequences). We made this point clear in the Methods part. Line 527 ~ 528:

“The orders of offers participants experienced were fully randomized within each experiment phase.”

STATISTICAL ANALYSIS & COMPUTATIONAL MODELING(4) In Study 1 DI offer types (90:10, 70:30), the rejection rate for DI-AI averse looks consistently higher than that for DI averse (ie, the blue line is above the yellow line). Is this significant? If so, how come? Since this is a between-subject design, I would not anticipate such a result (especially for the baseline). Also, for the LME results (eg, Table S3), only interactions were reported but not the main results.

We thank the Reviewer for pointing out this feature of the results. Prompted by this comment, we compared the baseline rejection rates between two conditions for these two offer types, finding in Experiment 1 that rejection rates in the DI-AI-averse condition were significantly higher than in the DI-averse condition (DI-AI-averse vs. DI-averse; Offer 90:10, β = 0.13, p < 0.001, Offer 70:30, β = 0.09, p < 0.034). We agree with the Reviewer that there should, in principle, be no difference between the experiences of participants in these two conditions is identical in the Baseline phase. However, we did not observe these difference in baseline preferences in Experiment 2 (DI-AI-averse vs. DI-averse; Offer 90:10, β = 0.07, p < 0.100, Offer 70:30, β = 0.05, p < 0.193). On the basis of the inconsistency of this effect across studies we believe this is a spurious difference in preferences stemming from chance.

Regarding the LME results, the reason why only interaction terms are reported is due to the specification of the model and the rationale for testing.

Taking the model reported in Table S3 as an example—a logistic model which examines Baseline phase rejection rates as a function of offer level and condition—the between-subject conditions (DI-averse and DI-AI-averse) are represented by dummy-coded variables. Similarly, offer types were also dummy-coded, such that each of the five columns (90:10, 70:30, 50:50, 30:70, and 10:90) correspond corresponded to a particular offer type. This model specification yields ten interaction terms (i.e., fixed effects) of interest—for example, the “DI-averse × Offer 90:10” indicates baseline rejection rates for 90:10 offers in DI-averse condition. Thus, to compare rejection rates across specific offer types, we estimate and report linear contrasts between these resultant terms. We have clarified the nature of these reported tests in our revised Results—for example, line189-190: “linear contrasts; e.g. 90:10 vs 10:90, all Ps<0.001, see Table S3 for logistic regression coefficients for rejection rates.

Also in response to this comment that and a recommendation from Reviewer 2 (see below), we have revised our supplementary materials to make each model specification clearer as SI line 25:

“*RejectionRate ~ 0 + (Disl + Advl):(Offer10 + Offer30 + Offer50 + Offer70 + Offer90)* + (1|*Subject*)”

(5) I do not particularly find this analysis appealing: "we examined whether participants' changes in rejection rates between Transfer and Baseline, could be explained by the degree to which they vicariously learned, defined as the change in punishment rates between the first and last 5 trials of the Learning phase." Naturally, the participants' behavior in the first 5 trials in the learning phase will be similar to those in the baseline; and their behavior in the last 5 trials in the learning phase would echo those at the transfer phase. I think it would be stronger to link the preference learning results to the change between the baseline and transfer phase, eg, by looking at the difference between alpha (beta) at the end of the learning phase and the initial alpha (beta).

Thanks for pointing this out. Also, considering the comments from Reviewer 2 concerning the interpretation of this analysis, we have elected to remove this result from our revision.

(6) I wonder if data from the baseline and transfer phases can also be modeled, using a simple Fehr-Schimdt model. This way, the change in alpha/beta can also be examined between the baseline and transfer phase.

We agree with the Reviewer that a simplified F-S model could be used, in principle, to characterize Baseline and Transfer phase behavior, but it is our view that the rejection rates provide readers with the clearest (and simplest) picture of how participants are responding to inequity. Put another way, we believe that the added complexity of using (and explaining) a new model to characterize simple, steady-state choice behavior (within these phases) would not be justified or add appreciable insights about participants’ behavior.

(7) I quite liked Study 2 which tests the generalization effect, and I expected to see an adapted computational modeling to directly reflect this idea. Indeed, the authors wrote, "[...] given that this model [...] assumes the sort of generalization of preferences between offer types [...]". But where exactly did the preference learning model assume the generalization? In the methods, the modeling seems to be only about Study 1; did the authors advise their model to accommodate Study 2? The authors also ran simulation for the learning phase in Study 2 (Figure 6), and how did the preference update (if at all) for offers (90:10 and 10:90) where feedback was not given? Extending/Unpacking the computational modeling results for Study 2 will be very helpful for the paper.

We are appreciative of the Reviewer’s positive impression of Experiment 2. Upon reflection, we realize that our original submission was not clear about the modeling done in Experiment 2, and we should clarify here that we did also fit the Preference Inference model to this dataset. As in Experiment 1, this model assumes that the participants have a representation of the teacher’s preference as a Fehr-Schmidt form utility function and infer the Teacher’s Envy and Guilt parameters through learning. The model indicates that, on the basis of experience with the Teacher’s preferences on moderately unfair offers (i.e., offer 70:30 and offer 30:70), participants can successfully infer these guess of these two parameters, and in turn, compute Fehr-Schmidt utility to guide their decisions in the extreme unfair offers (i.e., offer 90:10 and offer 10:90).

In response to this comment, we have made this clearer in our Results (Line 377-382):

“Finally, following Experiment 1, we fit a series of computational models of Learning phase choice behavior, comparing the goodness-of-fit of the four best-fitting models from Experiment 1 (see Methods). As before, we found that the Preference Inference model provided the best fit of participants’ Learning Phase behavior (Figure S1a, Table S12). Given that this model is able to infer the Teacher’s underlying inequity-averse preferences (rather than learns offer-specific rejection preferences), it is unsurprising that this model best describes the generalization behavior observed in Experiment 2.”

and in our revised Methods (Line 551-553)

“We considered 6 computational models of Learning Phase choice behavior, which we fit to individual participants’ observed sequences of choices, in both Experiments 1 and 2, via Maximum Likelihood Estimation”

**Reviewer #2 (Public review):**
Summary:This study investigates whether individuals can learn to adopt egalitarian norms that incur a personal monetary cost, such as rejecting offers that benefit them more than the giver (advantageous inequitable offers). While these behaviors are uncommon, two experiments demonstrate that individuals can learn to reject such offers through vicarious learning - by observing and acting in line with a "teacher" who follows these norms. The authors use computational modelling to argue that learners adopt these norms through a sophisticated process, inferring the latent structure of the teacher's preferences, akin to theory of mind.Strengths:This paper is well-written and tackles a critical topic relevant to social norms, morality, and justice. The findings, which show that individuals can adopt just and fair norms even at a personal cost, are promising. The study is well-situated in the literature, with clever experimental design and a computational approach that may offer insights into latent cognitive processes. Findings have potential implications for policymakers.Weaknesses:Note: in the text below, the "teacher" will refer to the agent from which a participant presumably receives feedback during the learning phase.(1) Focus on Disadvantageous Inequity (DI): A significant portion of the paper focuses on responses to Disadvantageous Inequitable (DI) offers, which is confusing given the study's primary aim is to examine learning in response to Advantageous Inequitable (AI) offers. The inclusion of DI offers is not well-justified and distracts from the main focus. Furthermore, the experimental design seems, in principle, inadequate to test for the learning effects of DI offers. Because both teaching regimes considered were identical for DI offers the paradigm lacks a control condition to test for learning effects related to these offers. I can't see how an increase in rejection of DI offers (e.g., between baseline and generalization) can be interpreted as speaking to learning. There are various other potential reasons for an increase in rejection of DI offers even if individuals learn nothing from learning (e.g. if envy builds up during the experiment as one encounters more instances of disadvantageous fairness).

We are appreciative of the Reviewer’s insight here and for the opportunity to clarify our experimental logic. We included DI offers in order to (1) expose participants to the full spectrum of offer types, and avoid focusing participants exclusively upon AI offers, which might result in a demand characteristic and (2) to afford exploration of how learning dynamics might differ in DI context s—which was, to some extent, examined in our previous study (FeldmanHall, Otto, & Phelps, 2018)—versus AI contexts. Furthermore, as this work builds critically on our previous study, we reasoned that replicating these original findings (in the DI context) would be important for demonstrating the generality of the learning effects in the DI context across experimental settings. We now remark on this point in our revised Introduction Line 129 ~132:

“In addition, to mechanistically probe how punitive preferences are acquired in Adv-I and Dis-I contexts—in turn, assessing the replicability of our earlier study investigating punitive preference acquisition in the Dis context—we also characterize trial-by-trial acquisition of punitive behavior with computational models of choice.”

(2) Statistical Analysis: The analysis of the learning effects of AI offers is not fully convincing. The authors analyse changes in rejection rates within each learning condition rather than directly comparing the two. Finding a significant effect in one condition but not the other does not demonstrate that the learning regime is driving the effect. A direct comparison between conditions is necessary for establishing that there is a causal role for the learning regime.

We agree with the Reviewer and upon reflection, believe that direct comparisons between conditions would be helpful to support the claim that the different learning conditions are responsible for the observed learning effects. In brief, these specific tests buttress the idea that exposure to AI-averse preferences result in increases in AI punishment rates in the Transfer phase (over and above the rates observed for participants who were only exposed to DI-averse preferences).

Accordingly, our revision now reports statistics concerning the differences between conditions for AI offers in Experiment 1 (Line 198~ 207):

“Importantly, when comparing these changes between the two learning conditions, we observed significant differences in rejection rates for Adv-I offers: compared to exposure to a Teacher who rejected only Dis-I offers, participants exposed to a Teacher who rejected both Dis-I and Adv-I offers were more likely to reject Adv-I offers and rated these offers more unfair. This difference between conditions was evident in both 30:70 offers (Rejection rates: β(SE) = 0.10(0.04), p = 0.013; Fairness ratings: β(SE) = -0.86(0.17), p < 0.001) and 10:90 offers (Rejection rates: β(SE) = 0.15(0.04), p < 0.001, Fairness ratings: β(SE) = -1.04(0.17), p < 0.001). As a control, we also compared rejection rates and fairness rating changes between conditions in Dis-I offers (90:10 and 30:70) and Fair offers (i.e., 50:50) but observed no significant difference (all ps > 0.217), suggesting that observing an Adv-I-averse Teacher’s preferences did not influence participants’ behavior in response to Dis-I offers.”

Line 222 ~ 230:

“A mixed-effects logistic regression revealed a significant larger (positive) effect of trial number on rejection rates of Adv-I offers for the Adv-Dis-I-Averse condition compared to the Dis-I-Averse condition. This relative rejection rate increase was evident both in 30:70 offers (Table S7; β(SE) = -0.77(0.24), p < 0.001) and in 10:90 offers (β(SE) = -1.10(0.33), p < 0.001). In contrast, comparing Dis-I and Fairness offers when the Teacher showed the same tendency to reject, we found no significant difference between the two conditions (90:10 splits: β(SE)=-0.48(0.21),p=0.593;70:30 splits: β(SE)=-0.01(0.14),p=0.150; 50:50 splits: β(SE)=-0.00(0.21),p=0.086). In other words, participants by and large appeared to adjust their rejection choices in accordance with the Teacher’s feedback in an incremental fashion.”

And in Experiment 2 Line 333 ~ 345:

“Similar to what we observed in Experiment 1 (Figure 4a), Compared to the participants in the Dis-I-Averse Condition, participants in the Adv-I-Averse Condition increased their rates of rejection of extreme Adv-I offerers (i.e., 10:90) in the Transfer Phase, relative to the Baseline phase (β(SE) = -0.12(0.04), p < 0.004; Table S9), suggesting that participants’ learned (and adopted) Adv-I-averse preferences, generalized from one specific offer type (30:70) to an offer types for which they received no Teacher feedback (10:90). Examining extreme Dis-I offers where the Teacher exhibited identical preferences across the two learning conditions, we found no difference in the Changes of Rejection Rates from Baseline to Transfer phase between conditions (β(SE) = -0.05(0.04), p < 0.259). Mirroring the observed rejection rates (Figure 4b), relative to the Dis-I-Averse Condition, participants’ fairness ratings for extreme Adv-I offers increased more from the Baseline to Transfer phase in the Adv-Dis-I-Averse Condition than in the Dis-I-Averse condition (β(SE) = -0.97(0.18), p < 0.001), but, importantly, changes in fairness ratings for extreme Dis-I offers did not differ significantly between learning conditions (β(SE) = -0.06(0.18), p < 0.723)”

Line 361 ~ 368:

“Examining the time course of rejection rates in Adv-I-contexts during the Learning phase (Figure 5) revealed that participants learned over time to punish mildly unfair 30:70 offers, and these punishment preferences generalized to more extreme offers (10:90). Specifically, compared to the Dis-I-Averse Condition, in the Adv-Dis-I-Averse condition we observed a significant larger trend of increase in rejections rates for 10:90 (Adv-I) offers (Figure 5, β(SE) = -0.81(0.26), p < 0.002 mixed-effects logistic regression, see Table S10). Again, when comparing the rejection rate increase in the extremely Dis-I offers (90:10), we didn’t find significant difference between conditions (β(SE) = -0.25(0.19), p < 0.707).”

(3) Correlation Between Learning and Contagion Effects:

The authors argue that correlations between learning effects (changes in rejection rates during the learning phase) and contagion effects (changes between the generalization and baseline phases) support the idea that individuals who are better aligning their preferences with the teacher also give more consideration to the teacher's preferences later during generalization phase. This interpretation is not convincing. Such correlations could emerge even in the absence of learning, driven by temporal trends like increasing guilt or envy (or even by slow temporal fluctuations in these processes) on behalf of self or others. The reason is that the baseline phase is temporally closer to the beginning of the learning phase whereas the generalization phase is temporally closer to the end of the learning phase. Additionally, the interpretation of these effects seems flawed, as changes in rejection rates do not necessarily indicate closer alignment with the teacher's preferences. For example, if the teacher rejects an offer 75% of the time then a positive 5% learning effect may imply better matching the teacher if it reflects an increase in rejection rate from 65% to 70%, but it implies divergence from the teacher if it reflects an increase from 85% to 90%. For similar reasons, it is not clear that the contagion effects reflect how much a teacher's preferences are taken into account during generalization.

This comment is very similar to a previous comment made by Reviewer 1, who also called into question the interpretability of these correlations. In response to both of these comments we have elected to remove these analyses from our revision.

(4) Modeling Efforts: The modelling approach is underdeveloped. The identification of the "best model" lacks transparency, as no model-recovery results are provided, and fits for the losing models are not shown, leaving readers in the dark about where these models fail. Moreover, the reinforcement learning (RL) models used are overly simplistic, treating actions as independent when they are likely inversely related (for example, the feedback that the teacher would have rejected an offer provides feedback that rejection is "correct" but also that acceptance is "an error", and the later is not incorporated into the modelling). It is unclear if and to what extent this limits current RL formulations. There are also potentially important missing details about the models. Can the authors justify/explain the reasoning behind including these variants they consider? What are the initial Q-values? If these are not free parameters what are their values?

We are appreciative of the Reviewer for identifying these potentially unaddressed questions.

The RL models we consider in the present study are naïve models which, in our previous study (FeldmanHall, Otto, & Phelps, 2018), we found to capture important aspects of learning. While simplistic, we believed these models serve as a reasonable baseline for evaluating more complex models, such as the Preference Inference model. We have made this point more explicit in our revised Introduction, Line 129 ~ 132:

“In addition, to mechanistically probe how punitive preferences may be acquired in Adv-I and Dis-I contexts—in turn, assessing the replicability of our earlier study investigating punitive preference acquisition in the Dis-I context—we also characterize trial-by-trial acquisition of punitive behavior with computational models of choice.”

Again, following from our previous modeling of observational learning (FeldmanHall et al., 2018), we believe that the feedback the Teacher provides here is ideally suited to the RL formalism. In particular, when the teacher indicates that the participant’s choice is what they would have preferred, the model receives a reward of ‘1’ (e.g., the participant rejects and the Teacher indicates they would preferred rejection, resulting in a positive prediction error) otherwise, the model receives a reward of ‘0’ (e.g., the participant accepts and the Teacher indicates they would preferred rejection, resulting in a negative prediction error), indicating that the participant did not choose in accordance with the Teacher’s preferences. Through an error driven learning process, these models provide a naïve way of learning to act in accordance with the Teacher’s preferences.

Regarding the requested model details: When treating the initial values as free parameters (model 5), we set Q(reject, offertype) as free values in [0,1] and Q(accept,offertype) as 0.5. This setting can capture participants' initial tendency to reject or accept offers from this offer type. When the initial values are fixed, for all offer types we set Q(reject, offertype) = Q(accept,offertype) = 0.5. In practice, when the initial values are fixed, setting them to 0.5 or 0 doesn’t make much difference. We have clarified these points in our revised Methods, Line 275 ~ 576:

“We kept the initial values fixed in this model, that is *Q*_0_(*reject,offertype*) = 0.5, (*offertype* ∈ 90:10, 70:30, 50:50, 30:70, 10:90)”

And Line 582 ~ 584:

“Formally, this model treats *Q*_0_(*reject,offertype*) = 0.5, (*offertype* ∈ 90:10, 70:30, 50:50, 30:70, 10:90) as free parameters with values between 0 and 1.”

(5) Conceptual Leap in Modeling Interpretation: The distinction between simple RL models and preference-inference models seems to hinge on the ability to generalize learning from one offer to another. Whereas in the RL models learning occurs independently for each offer (hence to cross-offer generalization), preference inference allows for generalization between different offers. However, the paper does not explore RL models that allow generalization based on the similarity of features of the offers (e.g., payment for the receiver, payment for the offer-giver, who benefits more). Such models are more parsimonious and could explain the results without invoking a theory of mind or any modelling of the teacher. In such model versions, a learner learns a functional form that allows to predict the teacher's feedback based on said offer features (e.g., linear or quadratic form). Because feedback for an offer modulates the parameters of this function (feature weights) generalization occurs without necessarily evoking any sophisticated model of the other person. This leaves open the possibility that RL models could perform just as well or even show superiority over the preference learning model, casting doubt on the authors' conclusions. Of note: even the behaviourists knew that as Little Albert was taught to fear rats, this fear generalized to rabbits. This could occur simply because rabbits are somewhat similar to rats. But this doesn't mean little Alfred had a sophisticated model of animals he used to infer how they behave.

We are appreciative of the Reviewer for their suggestion of an alternative explanation for the observed generalization effects. Our understanding of the suggestion, put simply, put simply, is that an RL model could capture the observed generalization effects if the model were to learn and update a functional form of the Teacher’s rejection preferences using an RL-like algorithm. This idea is similar, conceptually to our account of preference learning whereby the learner has a representation of the teacher’s preferences. In our experiment the offer is in the range of [0-100], the crux of this idea is why the participants should take the functional form (either v-shaped or quadratic) with the minimum at 50. This is important because, at the beginning of the learning phase, the rejection rates are already v-shaped with 50 as its minimum. The participants do not need to adjust the minimum of this functional form. Thus, if we assume that the participants represent the teacher’s rejection rate as a v-shape function with a minimum at [50,50], then this very likely implies that the participants have a representation that the teacher has a preference for fairness. Above all, we agree that with suitable setup of the functional form, one could implement an RL model to capture the generalization effects, without presupposing an internal “model” of the teacher’s preferences.

However, there is another way of modeling the generalization effect by truly “model-free” similarity-based Reinforcement learning. In this approach, we do not assume any particular functional form of the teacher’s preferences, but rather, assumes that experience acquired in one offer type can be generalized to offers that are close (i.e., similar) to the original offer. Accordingly, we implement this idea using a simple RL model in which the action values for each offer type is updated by a learning rate that is scaled by the distance between that offer and the experienced offer (i.e., the offer that generated the prediction error). This learning rate is governed by a Gaussian distribution, similar to the case in the Gaussian process regression (cf. Chulz, Speekenbrink, & Krause, 2018). The initial value of the ‘Reject’ action, for each offer , is set to a free parameter between 0 and 1, and the initial value for the 'Accept’ action was set to 0.5. The results show that even though this model exhibits the trend of increasing rejection rates observed in the AI-DI punish condition, the initial preferences (i.e., starting point of learning) diverges markedly from the Learning phase behavior we observed in Experiment 1:

This demonstrated that the participant at least maintains a representation of the teacher’s preference at the beginning. That is, they have prior knowledge about the shape of this preference. We incorporated this property into the model, that is, we considered a new model that assumes v-shaped starting values for rejection with two parameters, alpha and beta, governing the slope of this v-shaped function (this starting value actually mimics the shape of the preference functions of the Fehr-Schmidt model). We found that this new model (which we term the “Model RL Sim Vstart”) provided a satisfactory qualitative fit of the Transfer phase learning curves in Experiment 1 (see below).

**Author response image 2. sa3fig2:** 

However, we didn’t adopt this model as the best model for the following reasons. First, this model yielded a larger AIC value (indicating worse quantitative fit) compared to our preference Inference model in both Experiments 1 and 2, likely owing to its increased complexity (5 free parameters versus 4 in the Preference Inference model). Accordingly, we believe that inclusion of this model in our revised submission would be more distracting than helpful on account of the added complexity of explaining and justifying these assumptions, and of course its comparatively poor goodness of fit (relative to the preference inference model).

(6) Limitations of the Preference-Inference Model: The preference-inference model struggles to capture key aspects of the data, such as the increase in rejection rates for 70:30 DI offers during the learning phase (e.g. Figure 3A, AI+DI blue group). This is puzzling.Thinking about this I realized the model makes quite strong unintuitive predictions that are not examined. For example, if a subject begins the learning phase rejecting the 70:30 offer more than 50% of the time (meaning the starting guilt parameter is higher than 1.5), then overleaning the tendency to reject will decrease to below 50% (the guilt parameter will be pulled down below 1.5). This is despite the fact the teacher rejects 75% of the offers. In other words, as learning continues learners will diverge from the teacher. On the other hand, if a participant begins learning to tend to accept this offer (guilt < 1.5) then during learning they can increase their rejection rate but never above 50%. Thus one can never fully converge on the teacher. I think this relates to the model's failure in accounting for the pattern mentioned above. I wonder if individuals actually abide by these strict predictions. In any case, these issues raise questions about the validity of the model as a representation of how individuals learn to align with a teacher's preferences (given that the model doesn't really allow for such an alignment).

In response to this comment we explain our efforts to build a new model that might be able conceptually resolves the issue identified by the Reviewer.

The key intuition guiding the Preference inference model is a Bayesian account of learning which we aimed to further simplify. In this setting, a Bayesian learner maintains a representation of the teacher’s inequity aversion parameters and updates it according to the teacher’s (observed) behavior. Intuitively, the posterior distribution shifts to the likelihood of the teacher’s action. On this view, when the teacher rejects, for instance, an AI offer, the learner should assign a higher probability to larger values of the Guilt parameter, and in turn the learner should change their posterior estimate to better capture the teacher’s preferences.

In the current study, we simplified this idea, implementing this sort of learning using incremental “delta rule” updating (e.g. Equation 8 of the main text). Then the key question is to define the “teaching signal”. Assuming that the teacher rejects an offer 70:30, based on Bayesian reasoning, the teacher’s envy parameter (α) is more likely to exceed 1.5 (computed as 30/(50-30), per equation 7) than to be smaller than 1.5. Thus, 1.5, which is then used in equation 8 to update α, can be thought of as a teaching signal. We simply assumed that if the initial estimate is already greater than 1.5, which means the prior is consistent with the likelihood, no updating would occur. This assumption raises the question of how to set the learning rate range. In principle, an envy parameter that is larger than 1.5 should be the target of learning (i.e., the teaching signal), and thus our model definition allows the learning rate to be greater than 1, incorporating this possibility.

Our simplified preference inference model has already successfully captured some key aspects of the participants’ learning behavior. However, it may fail in the following case: assume that the participant has an initial estimate of 1.51 for the envy parameter (β). Let’s say this corresponds to a rejection rate of 60%. Thus, no matter how many times the teacher rejects the offer 70:30, the participant’s estimate of the envy parameter remains the same, but observing only one offer acceptance would decrease this estimate, and in turn, would decrease the model’s predicted rejection rate. We believe this is the anomalous behavior—in 70:30 offers—identified by the Reviewer which the model does not appear able to recreate participants’ in these offers.

This issue actually touches the core of our model specification, that is, the choosing of the teaching signal. As we chose 1.5 as the teaching signal—i.e. lower bound on whenever the teacher rejects or accepts an offer of 70:30, a very small deviation of 1.5 would fail one part of updating. One way to mitigate this problem would be to choose a lower bound for α greater than 1.5, such that when the Teacher rejects a 70:30 offer, we assign a number greater than 1.5 (by ‘hard-coding’ this into the model via modification of equation 7). One sensible candidate value could be the middle point between 1.5 and 10 (the maximum value of α per our model definition). Intuitively, the model of this setting could still pull up the value of α to 1.51 when the teacher rejects 70:30, thus alleviating (but not completely eliminating) the anomaly.

We fitted this modified Preference Inference model to the data from Experiment 1 (see Author response image 3 below) and found that even though this model has a smaller AIC (and thus better quantitative fit than the original Preference Inference model), it still doesn’t fully capture the participants’ behavior for 70:30 offers.

**Author response image 3. sa3fig3:** 

Accordingly, rather than revising our model to include an unprincipled ‘kludge’ to account for this minor anomaly in the model behavior, we have opted to report our original model in our revision as we still believe it parsimoniously captures our intuitions about preference learning and provides a better fit to the observed behavior than the other RL models considered in the present study.

**Reviewer #1 (Recommendations for the authors):**
(1) I do not particularly prefer the acronyms AI and DI for disadvantageous inequity and advantageous inequity. Although they have been used in the literature, not every single paper uses them. More importantly, AI these days has such a strong meaning of artificial intelligence, so when I was reading this, I'd need to very actively inhibit this interpretation. I believe for the readability for a wider readership of eLife, I would advise not to use AI/DI here, but rather use the full terms.

We thank the Reviewer for this suggestion. As the full spelling of the two terms are somewhat lengthy, and appear frequently in the figures, we have elected to change the abbreviations for disadvantageous inequity and advantageous inequity to Dis-I and Adv-I, respectively in the main text and the supplementary information. We still use AI/DI in the response letter to make the terminology consistent.

(2) Do "punishment rate" and "rejection rate" mean the same? If so, it would be helpful to stick with one single term, eg, rejection rate.

We thank the Reviewer for this suggestion. As these terms have the same meaning, we have opted to use the term “rejection rate” throughout the main text.

(3) For the linear mixed effect models, were other random effect structures also considered (eg, random slops of experimental conditions)? It might be worth considering a few model specifications and selecting the best one to explain the data.

Thanks for this comment. Following established best practices (Barr, Levy, Scheepers, & Tily, 2013) we have elected to use a maximal random effects structure, whereby all possible predictor variables in the fixed effects structure also appear in the random effects structure.

(4) For equation (4), the softmax temperature is denoted as tau, but later in the text, it is called gamma. Please make it consistent.

We are appreciative of the Reviewer’s attention to detail. We have corrected this error.

**Reviewer #2 (Recommendations for the authors):**
(1) Several Tables in SI are unclear. I wasn't clear if these report raw probabilities of coefficients of mixed models. For any mixed models, it would help to give the model specification (e.g., Walkins form) and explain how variables were coded.

We are appreciative of the Reviewer’s attention to detail. We have clarified, in the captions accompanying our supplemental regression tables, that these coefficients represent log-odds. Regretfully we are unaware of the “Walkins form” the Reviewer references (even after extensive searching of the scientific literature). However, in our new revision we do include lme4 model syntax in our supplemental information which we believe will be helpful for readers seeking replicate our model specification.

(2) In one of the models it was said that the guilt and envy parameters were bounded between 0-1 but this doesn't make sense and I think values outside this range were later reported.

We are again appreciative of the Reviewer’s attention to detail. This was an error we have corrected— the actual range is [0,10].

(3) It is unclear if the model parameters are recoverable.

In response to this comment our revision now reports a basic parameter recovery analysis for the winning Preference Inference model. This is reported in our revised Methods:

“Finally, to verify if the free parameters of the winning model (Preference Inference) are recoverable, we simulated 200 artificial subjects, based on the Learning Phase of Experiment 1, with free parameters randomly chosen (uniformly) from their defined ranges. We then employed the same model-fitting procedure as described above to estimate these parameter value, observing that parameters. We found that all parameters of the model can be recovered (see Figure S2).”

And scatter plots depicting these simulated (versus recovered) parameters are given in Figure S2 of our revised Supplementary Information:

(4) I was confused about what Figure S2 shows. The text says this is about correlating contagious effects for different offers but the captions speak about learning effects. This is an important aspect which is unclear.

We have removed this figure in response to both Reviewers’ comments about the limited insights that can be drawn on the basis of these correlations.